# Flash Flood Events along the West Mediterranean Coasts: Inundations of Urbanized Areas Conditioned by Anthropic Impacts

**Francesco Faccini [1,2], Fabio Luino [2,*], Guido Paliaga [2], Anna Roccati [2] and Laura Turconi [2]**

1 Department of Earth, Environmental and Life Sciences, University of Genoa, Corso Europa 26, 16132 Genoa, Italy; faccini@unige.it

2 National Research Council, Research Institute for Geo-Hydrological Protection, Strada delle Cacce 73, 10135 Turin, Italy; guido.paliaga@irpi.cnr.it (G.P.); anna.roccati@irpi.cnr.it (A.R.); laura.turconi@irpi.cnr.it (L.T.)

* Correspondence: fabio.luino@irpi.cnr.it; Tel.: +39-011-397-7823

**Abstract:** Flash floods represent one of the natural hazards that causes the greatest number of victims in the Mediterranean area. These processes occur by short and intense rainfall affecting limited areas of a few square kilometers, with rapid hydrological responses. Among the causes of the flood frequency increase in the last decades are the effects of the urban expansion in areas of fluvial pertinence and climatic change, namely the interaction between anthropogenic landforms and hydro-geomorphological dynamics. In this paper the authors show a comparison between flood events with very similar weather-hydrological characteristics and the ground effects occurred in coastal areas of three regions located at the top of a triangle in the Ligurian Sea, namely Liguria, Tuscany and Sardinia. With respect to the meteorological-hydrological hazard, it should be noted that the events analyzed occurred during autumn, in the conditions of a storm system triggered by cyclogenesis on the Genoa Gulf or by the extra-tropical cyclone Cleopatra. The "flash floods" damage recorded in the inhabited areas is due to the vulnerability of the elements at risk in the fluvio-coastal plains examined. There are numerous anthropogenic forcings that have influenced the hydro-geomorphological dynamics and that have led to an increase in risk conditions.

**Keywords:** flash floods; intense rainfall; urbanized areas; damage; anthropic impacts; West Mediterranean; Italy

## 1. Introduction

Climatic changes, including the increase in extreme temperatures and number of heavy rainfall events, have been recorded since around 1950 in different areas of the world [1]. Floods are already the most frequent in European coastal areas [2–4] and among the costliest and deadliest natural processes in the Mediterranean area [5].

Based on an international disaster database [6], 200 billion Euros in damages related to various calamities since 1900 have occurred in the countries surrounding the Mediterranean Sea out of which 85 billion are related to river flooding [7,8].

The observed variability of flood frequency and discharge magnitude is therefore the result of a complex interaction between rainfall history and the factors that control the response of the river basins, in particular, run-off modes. In the Mediterranean environment, the global warming process manifests itself with an increase in the average annual air temperature and with a variation in the rainfall regime [9,10]. Some regions such as Liguria, Tuscany and Sardinia in Italy; Provence-Alpes-Cote d'Azur and Corse in France; and Catalonia and the Valencia area in Spain are particularly exposed to flash floods [7,11] for which rainfall peaks and the flood peaks of the watercourses are very close in time.

This particular pattern is the result of the interplay between the dominant atmospheric low level flow circulation patterns and the relief and orientations of the northern Mediterranean coasts, which forces convergence and triggers convection.

The magnitude and impact of extreme floods vary significantly over the Mediterranean region with a significative difference between the West and East parts [12]. The western part of the region is much more prone to high impact and high magnitude events [13–16]. This is probably due to: (a) The proximity of the Atlantic Ocean and oceanic climatic influences at latitudes where eastward atmospheric flows dominate [7,17]; (b) the reliefs surrounding the West Mediterranean Sea forces the convergence of low-level atmospheric flows and the uplift of warm wet air masses that drift from the sea to the coasts, thereby triggering active convection and consequently short intense bursts of rainfall.

The coasts of the Mediterranean Sea are characterized by short and intense rainfalls [13,17], which in recent decades have shown an increasing frequency [18,19]. Furthermore, for Italy, the statistics seem to indicate an increase in geo-hydrological phenomena in small basins for the last 30–50 years [20]. These rainfalls, especially if concentrated on reliefs not far from the coasts, can generate the phenomenon of flash floods along the short streambeds that have significant slopes [21–23]. Such precipitations are characterized by convective events, typically with 100 mm or more cumulated rainfalls over a few hours. The affected areas are often limited to a few hundred square kilometers, with rapid hydrological responses, e.g., less than 6 h delay between the peak rainfall intensity and peak water discharge downstream [8].

The coastal urban flooding is a complex phenomenon which may occur in various forms such as: Urban flooding due to high intensity rainfall (pluvial flooding); urban flooding due to inadequate drainage; flooding caused by overtopping in the channels or streams/rivers. In coastal urban cities such as Genoa, Olbia and Livorno, severe flood scenarios mostly take place due to combination of surface flooding and stream overtopping. Urban flooding is one of the most severe problem in numerous parts of the world because they affect goods and can cause casualties. Urban flood, being a natural disaster, cannot be avoided; however, the losses incurred due to flooding can be reduced by proper flood mitigation planning. As such, it is necessary to have a proper estimation of flood extent and flood risk for the different flow conditions so that proper flood evacuation and disaster management plan can be prepared in advance.

The flash flood consequences and the ground effects are amplified if the floodwaters spread to densely urbanized areas [24–27]. They are usually crossed by canalized streams that are often culverted for long stretches. These streambeds, which have been narrowed year after year to acquire new urban spaces [28,29], are often surmounted by bridges that are inadequate, with spans that are clearly insufficient for the discharge of flood waters.

Flash floods characterized by severe ground effects are generally triggered by:

1. Short-lived (often less than 3 h) strongly convective intense rainfall events, with total rainfall amounts (200–300 mm). Such violent events have a limited areal extent (<100 km$^2$) and generate local floods of small headwater streams that usually possess a surface of <40 km$^2$. A typical example of such flash floods is the catastrophic flash flood that occurred in eastern part of Genoa Metropolitan Area in November 2002 [30].

2. Mesoscale convective systems can produce stationary rainfall amounts exceeding 200–300 mm in a few hours [31]. During two severe cloudbursts that hit the Liguria coasts in the last 10 years, 539.0 mm/24 h were recorded at the Brugnato rain gauge station, during the famous Cinque Terre event, in Eastern Liguria during October 2011 [32] and 556.0 mm/24 h at Quezzi station, Bisagno Valley, Genoa city in November 2011 [33,34]. The areal extent of such events ranges from less than 100 km$^2$ to greater than hundreds of km$^2$.

3. On some occasions, heavy and prolonged rainfall may be part of a large-scale perturbation lasting several days. In such cases, extreme rainfall accumulation may be observed locally: 700 mm over 6 days (up to 1800 mm in October and November) caused floods and further loss of life on the border between Liguria and Piedmont

during the events of 21–22 October 2019 and 23–24 November 2019 [35]. These events generally cover a larger area from hundreds to thousands of km².

Along the Italian coasts, during the period from September to November, the so-called "Meteorological Fall" is the main season for flash floods that cause severe damage and often casualties due to their suddenness. This is particularly the case of mesoscale convective systems producing long lasting and stationary rainfall events that lead to strong responses by the watersheds concerned (i.e., high runoff rates due to soil saturation) and substantial coincidence between the peak rainfall and flood peak in small hydrographic basins (<250 km²).

The north-western coasts of Italy are historically subject to flash floods: Ligurian coasts, along with the Tuscan coasts and coasts in Sardinia. There are at least 46 damaging flood events that have been recorded in the last 30 years, practically one every 7.8 months (Table 1): 9 events occurred in Tuscany, while 19 cases occurred in Liguria and Sardinia.

**Table 1.** Severe meteorological events during the period 2000–2020 that occurred in Sardinia and along the coasts of Liguria and Tuscany with severe consequences (flooding/flash floods) for urban areas and inhabitants. The content of the brackets displays the victim numbers. The three events described in this article are bolded.

| Year | Month | Day | Place/Area (Casualties) | Region |
|------|-------|-----|-------------------------|--------|
| 2020 | Nov | 28 | Bitti (3) | Sardinia |
| 2020 | Oct | 2–3 | Ventimiglia (10) | Liguria |
| **2017** | **Sept** | **9–10** | **Livorno (8)** | **Tuscany** |
| 2015 | Jul | 22 | Cagliari | Sardinia |
| 2014 | Oct | 14 | Grosseto and Orbetello | Tuscany |
| 2014 | Nov | 5 | Carrara (1) | Tuscany |
| 2014 | Nov | 10 | Recco-Chiavari-Camogli (2) | Liguria |
| 2014 | Oct | 9–10 | Genoa (1) | Liguria |
| **2013** | **Nov** | **18** | **Olbia (18)** | **Sardinia** |
| 2012 | Nov | 28 | Carrara and Ortonovo | Tuscany |
| 2012 | Nov | 12 | Maremma and Grosseto (7) | Tuscany |
| 2011 | Nov | 7 | Isola d'Elba (1) | Tuscany |
| 2011 | Nov | 4 | Genoa (6) | Liguria |
| 2011 | Oct | 24–25 | Cinque Terre e Lunigiana (13) | Liguria and Tuscany |
| **2010** | **Oct** | **4** | **Genoa Sestri Ponente (1)** | **Liguria** |
| 2010 | Sept | 7 | Genoa | Liguria |
| 2010 | Jan | 25–26 | Olbia | Sardinia |
| 2008 | Nov | 4 | Olbia | Sardinia |
| 2008 | Oct | 22 | Capoterra (5) | Sardinia |
| 2008 | Jun | 16 | Genoa | Liguria |
| 2007 | Jun | 1 | Genoa | Liguria |
| 2006 | Nov | 13 | Cagliari | Sardinia |
| 2006 | Sept | 25–26 | Cagliari | Sardinia |
| 2006 | Sept | 15 | Bordighera and Vallecrosia | Liguria |
| 2006 | Aug | 16–17 | Genoa | Liguria |
| 2005 | Nov | 13 | Cagliari | Sardinia |
| 2005 | Apr | 5–6 | Cagliari, Capoterra, Pula | Sardinia |
| 2004 | Dec | 6 | Villanova Strisaili (2) | Sardinia |
| 2003 | Sept | 23 | Massa-Carrara (2) | Tuscany |
| 2002 | Nov | 23 | Imperia and Genoa | Liguria |
| 2002 | Nov | 11 | Cagliari | Sardinia |
| 2002 | Oct | 9 | Cagliari | Sardinia |
| 2000 | Nov | 6 | Imperia and Savona | Liguria |

On the one hand we can affirm that flash floods are geo-hydrological processes linked to particular hydro-meteorological conditions and that their behaviors are significantly affected by climate change [36], but on the other hand we cannot omit the fundamental role played by wild urbanization [37], which has affected many towns on the western Mediterranean coast. This uncontrolled expansion occurred after the Second World War and appears to have been particularly significant in the already notoriously hazardous areas that have undergone and imparted important changes to the hydrographic network over time.

There are some fundamental reasons that constitute the basis of the decision to write this paper. First of all, the fact that no author has so far compared the events that took place in these regions, which are located adjacent to one another along the coast around the Upper Tyrrhenian Sea, in the national and international body of literature and research. The second reason is to compare the dynamics of three similar flood events, both in terms of meteorological and hydrological characteristics, and in terms of effects on the ground, with particular attention to the identification of any anthropogenic factors. In detail, three flash flood events were chronologically analyzed (Figure 1): (1) Genoa Sestri Ponente (Liguria), which occurred on 4 October 2010; (2) Olbia flood (Sardinia) on 18 November 2013 triggered by cyclone Cleopatra; (3) Livorno flood (Tuscany) on 9 September 2017 triggered by the cyclone Genoa Low.

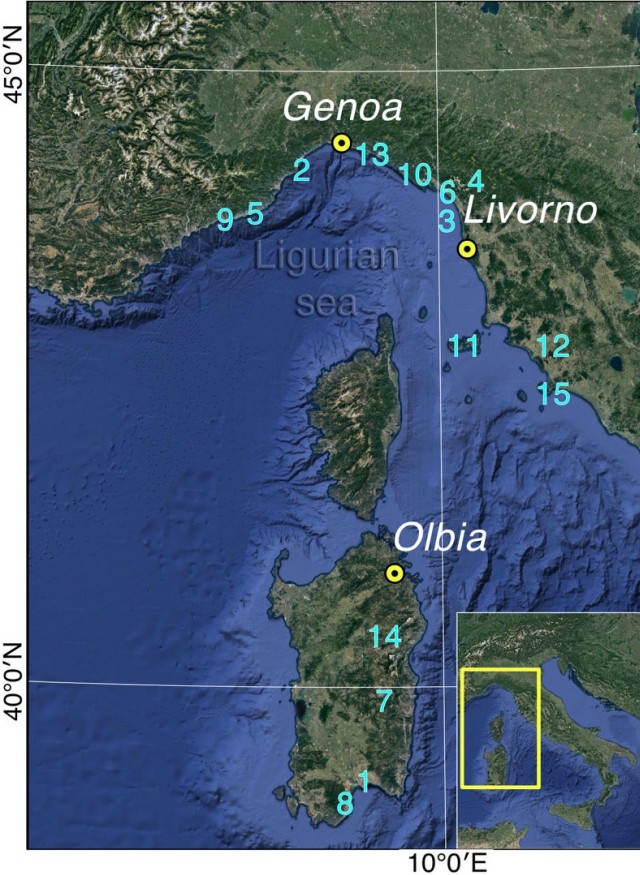

**Figure 1.** Geographic location of the case-studies: Genoa Sestri Ponente (Liguria), Livorno (Tuscany) and Olbia (Sassari, Sardinia). Other places named in the text: (1) Cagliari; (2) Savona; (3) Versilia; (4) Garfagnana; (5) Imperia; (6) Massa and Carrara; (7) Villanova Strisaili; (8) Capoterra; (9) Bordighera, Vallecrosia, Ventimiglia; (10) Cinque Terre; (11) Elba Island; (12) Grosseto, Maremma; (13) Recco, Camogli, Chiavari; (14) Bitti; (15) Orbetello.

## 2. Study Area

### 2.1. Genoa Sestri Ponente City

Sestri Ponente is a district of the Genoa Metropolitan City, the regional chief town of Liguria (northern Italy) (Figures 1 and 2A); it extends over an area of about 8 km², with a population of 45,000 inhabitants. Until 1926, Sestri Ponente was an autonomous municipality until it was incorporated, together with other municipalities of the Genoa neighborhood, called "genovesato", into the "Big Genova" City. The historic core of Sestri Ponente is represented by a narrow coastal strip, about 1 km long, running parallel to the shoreline and included in the floodplain between the Chiaravagna stream to the east and the Cantarena stream to the west. The industrial expansion in the 1920s and 1930s developed in the alluvial plain around the historic core, while the subsequent urban sprawl from the 1960s to date has continued upstream and occupies the hill slopes behind (Figure 2A). Now the city is developed for 2.6 km in length along the coastline.

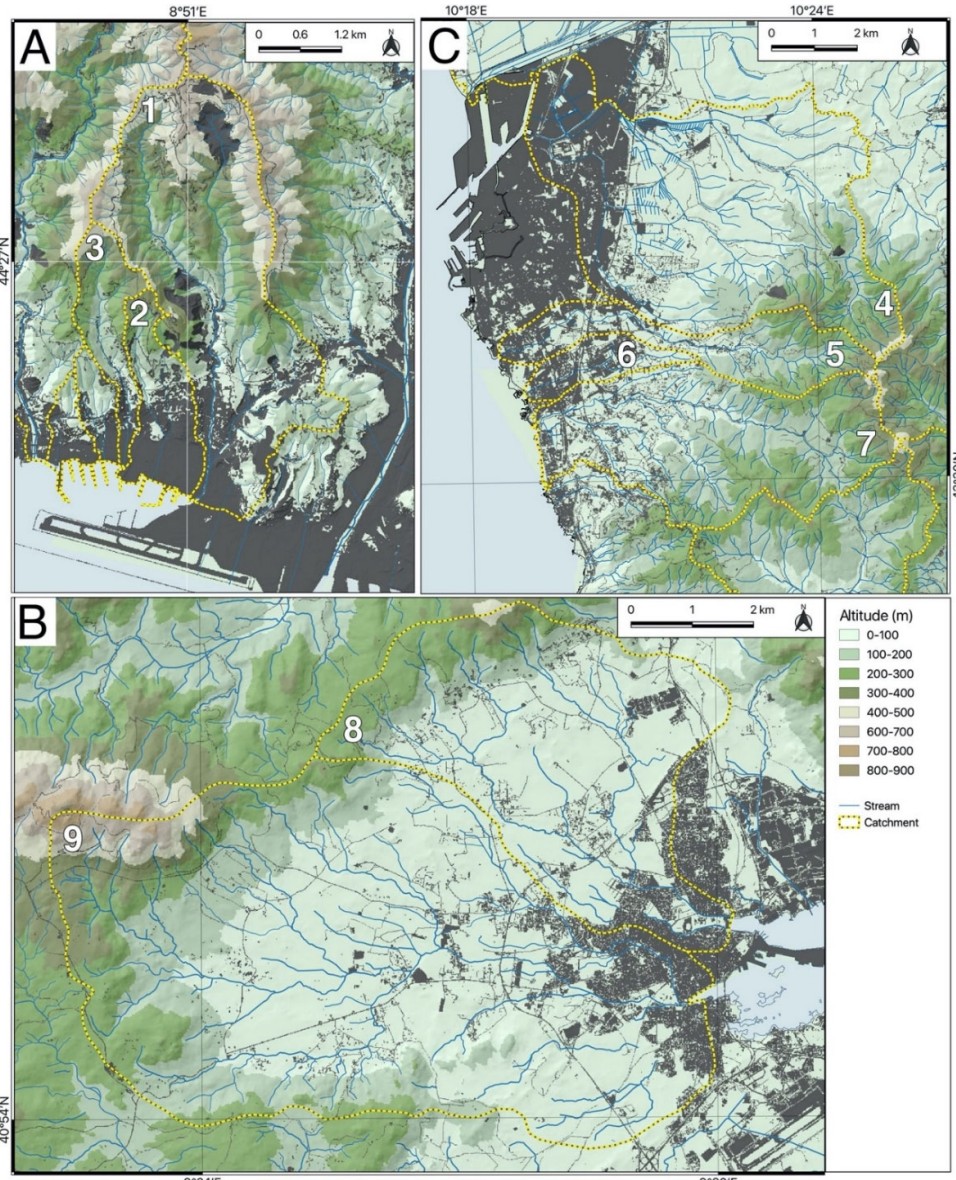

**Figure 2.** Elevation maps and catchments of the case studies: Genoa Sestri Ponente (**A**), Olbia (**B**) and Livorno (**C**). Numbers refer to catchments in Table 2.

From the geomorphological point of view, Sestri Ponente stands on a narrow alluvial plain that is 2 km length and less than 500 m wide and is genetically linked to the action of several watercourses and significantly modified by anthropogenic landforms. Drainage networks are well developed with a torrential hydrological regime. Catchments are very small in size (generally <10 km$^2$) and characterized by high energy relief (Table 2) due to elevation exceeding 700 m a.s.l. at a few kilometers from the coast and high slope gradient locally higher than 60% (Figure 3A): Rio Chiaravagna Stream (11 km$^2$, with an estimated maximum full-flow rate of 213 m$^3$/s for a 50-years return period), Rio Cantarena Stream (1.58 km$^2$, 52 m$^3$/s), Rio Molinassi Stream (2.00 km$^2$, 66 m$^3$/s) and Rio Marotto Creek (0.67 km$^2$, 22 m$^3$/s) [38]. The final stretches of these waterways are generally canalized and drained.

**Table 2.** Morphometric parameters of the basins analyzed (the numbers refer to Figure 2). H Max, maximum altitude; H Med, mean altitude; G med, mean gradient; Sup, surface; Ss, soil sealing; L rn, river network total length; Dd, Drainage density.

| Area | Catchment | H Max (m) | H Med (m) | G Med (%) | Sup (km$^2$) | Ss (%) | L Rn (km) | Dd (km$^{-1}$) |
|---|---|---|---|---|---|---|---|---|
| Genoa | 1 | 658 | 262 | 47.2 | 11 | 23.4 | 60.5 | 5.7 |
| | 2 | 438 | 89 | 31.8 | 1.9 | 49.4 | 5.6 | 3.0 |
| | 3 | 545 | 230 | 49.4 | 1.8 | 20.4 | 9.2 | 5.1 |
| Livorno | 4 | 430 | 71 | 13.4 | 37.4 | 23.9 | 126.0 | 3.4 |
| | 5 | 456 | 132 | 19 | 8.9 | 22.9 | 26.0 | 2.9 |
| | 6 | 107 | 26 | 3 | 3.0 | 57.2 | 5.4 | 1.8 |
| | 7 | 456 | 148 | 20.6 | 21.9 | 13.1 | 71.1 | 3.3 |
| Olbia | 8 | 457 | 89 | 10 | 20.9 | 14.1 | 37.0 | 1.8 |
| | 9 | 700 | 115 | 12.8 | 48.2 | 10.7 | 104.6 | 2.2 |

Land use in the studied catchments mainly consists of artificial surfaces (32.5%) in the lower part and forests and seminatural areas (45.0%) in the upper part of the basins; subordinately are agricultural areas (7.0%) (Supplementary Materials Table S1)

The climate of Sestri Ponente is typically Mediterranean, with hot summers and relatively mild winters: the average annual temperature is about 15 °C with a significant upward trend. The mean annual rainfall during the period from 1960 to 2019 is about 1100 mm and the annual rainy days are 71 with an average rainfall rate of 14 mm/d, which is also increasing [21]. The episodes of intense and short-lived rain are frequent, especially in the autumn months (October and November), when typically small-sized and quasi-stationary V-shaped convective systems are generated [31]. These systems result in extreme flood-causing rainfalls, for which ground effects are often disastrous. In the post-war period alone, flood events occurred 10 times in 69 years, which on average amounts to 1 event every 6.9 years (Supplementary Materials Table S2).

*2.2. Olbia City*

The Olbia city is one of the major cities of Sardinia and the most important link with the mainland and other Mediterranean countries due to its tourist and commercial port (Figures 1 and 2B). It extends for about 40 km$^2$ (including the more peripheral districts) with a population of about 60,000 inhabitants, while during the summer it can reach 100,000 inhabitants.

The whole coastal sector has undergone important changes since the early 1900s as a result of management and sanitation. In the first decades of the 20th century, large swampy and brackish areas were reclaimed. Then, beginning from the 1960s Olbia experienced a demographic increase related to the tourist development in the neighboring Costa Smeralda, which is world-famous for its sea landscape and resorts. In 50 years, the

town has tripled its population, transforming the socio-economic fabric and the population of inhabitants increased from 17,800 inhabitants (1961) to 60,000 today.

The plain of Olbia occupies a structural depression resulting from the kinematics of the Sardinian-Mediterranean block related to the formation of the Western Tyrrhenian Sea [39]. Due to the irregular and indented coastal morphology, with parallel hollow rias separated by prominent ridges featured by typical plateau and sierras, drainage patterns are mainly dendritic and slightly developed; the upper part of catchments locally exceeds 700 m a.s.l. and the watercourses that result are generally steep (Figure 3B). As a consequence of the progressive urbanization of the plain, streams crossing the urban area are strongly modified due to canalization and flow regulation. The main watercourses (Table 2) are the Rio Seligheddu Stream (38.4 km$^2$, with a maximum capacity of 330 m$^3$/s for a 200 year return period), the Rio San Nicola Stream (30 km$^2$, 170 m$^3$/s) and the Rio Gadduresu Stream (7 km$^2$, 55 m$^3$/s) as well as other minor ones (<5 km$^2$), resulting in an overall flow rate exceeding 600 m$^3$/s [40,41].

Land use in the studied catchments consists mainly in agricultural use (63.9%); artificial surfaces are present in suborder (12.9%) and forests and seminatural areas (22.7%) (Supplementary Materials Table S1).

The climate is typically Mediterranean, with mild and humid winters and hot and dry summers. The average annual temperature is about 16 °C. Sardinia is located almost at the center of a low-pressure area that determines the convergence of different air masses and the formation of self-regenerating convective systems, especially during the winter period, resulting in strong "V-shaped" marine thunderstorms. The mean annual precipitation ranges between 600 and 900 mm and the annual rainy days are 60 (rainfall rate 10–15 mm/d): rainfall is more abundant in the autumn and winter months (from October to December), while a minimum rainfall peak occurs in summer [41]. The urban development of Olbia is relatively recent. As a consequence, only documents of floods post Second World War have been discovered and considered: at least 16 flash floods in the last 69 years (on average 1 event every 4.3 years) (Supplementary Materials Table S2).

*2.3. Livorno City*

Livorno city extends for just over 100 km$^2$ of surface along the Tyrrhenian coast and represents one of the most important trade and industrial centers in Tuscany (central Italy) (Figures 1 and 2C): the urban area is subdivided into numerous districts and a total of approximately 158,000 inhabitants are counted. The city has already developed in historical times around the port area, with progressively more important enlargements since the beginning of 1900 linked to the development of communications and industrial activities. Urbanization has resulted in the growing occupation of natural drainage areas and floodplains [29,42]. Most expansions relate to the period after Second World War.

Livorno town stands on a flat sector (in the north-side and along the coast) which corresponds to a polycyclic marine terrace characterized by the presence of three orders of sea terraces that are, at least, aged between the middle Pleistocene and the upper Pleistocene. They are crossed by a hydrographic network that consists of several streams. The alluvial plain is, in fact, a low coastal terrace located north of the city and is shaped by the course of the River Arno in the mouth area.

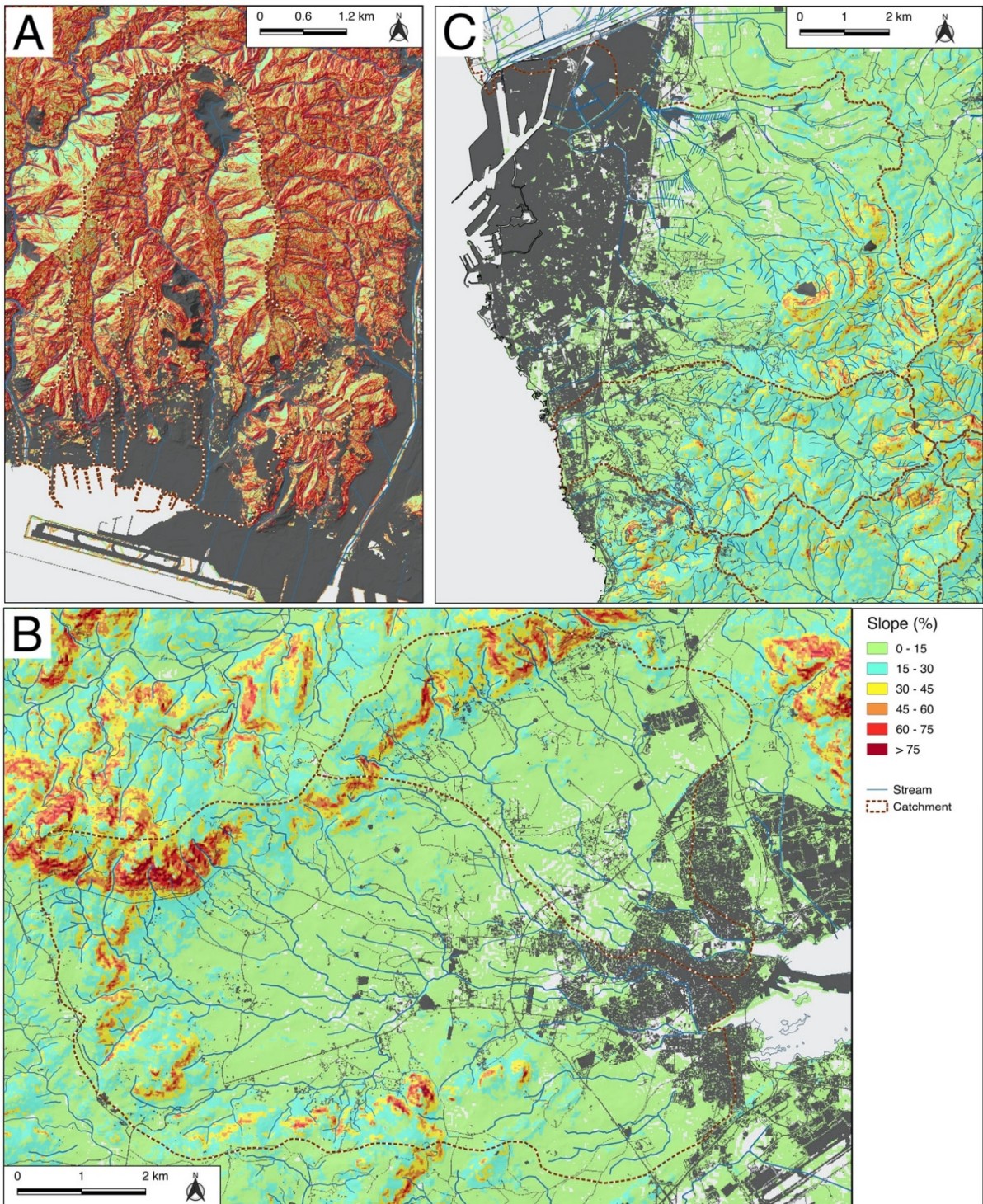

**Figure 3.** Slope maps and catchments of the case-studies: Genoa Sestri Ponente (**A**), Olbia (**B**) and Livorno (**C**).

Catchments are small (Table 2), with a size of generally <30 km²; they do not reach particularly high elevation (about 400 m a.s.l.) in their upper sectors; they possess mildly steep terrain and gentle hilly slopes (Figure 3C). Drainage networks are well developed; they possess watercourses of limited length; and a well-defined grid of small tributaries: the major ones are the Rio Ugione Stream (33.2 km² with a maximum evaluated discharge

of 137 m³/s for a 200-years return period), the Rio Ardenza Stream (21.2 km², 284 m³/s) and the Rio Maggiore Stream (8 km², 100 m³/s) [43].

Land use in the studied catchments mainly consists of forests and seminatural areas (50.0%); agricultural are present in suborder (28.5%) and artificial surfaces (20.4%) (Supplementary Materials Table S1).

During the period 1969–2018, the mean annual rainfall recorded in Livorno is about 800 mm that is mainly concentrated during the autumn, with an average of 74 rainy days (rainfall rate 10.7 mm/d) [44]. The climate is Mediterranean, characterized by warm summers, mitigated by the presence of sea breezes and not particularly cold winters. The average annual temperature is about 15.8 °C.

Damaging effects on the ground in Livorno are associated with both flooding of water streams and pluvial floods due to intense rainfall events: before the last event, they occurred at least 16 times in 71 years. On average there is 1 event every 4.4 years (Supplementary Materials Table S2).

## 3. Materials and Methods

### 3.1. Research Methodology

The applied methodology has been developed in different phases. First of all, a close bibliographic and cartographic research has been carried out, with the aim of cataloguing recent and historical floods that occurred in the three study areas (Genoa, Olbia and Livorno) and correlate regions. Flood and damage information have been derived from different sources: (i) newspapers articles and chronicles notes from local media, (ii) inedited documents and technical and event reports collected in archives of local municipalities, (iii) books and scientific papers gathered in the libraries and iv) interviews with local inhabitants.

We obtained rainfall and hydrological data about recent and past events from Hydrological Annals edited by SIMN ("National hydrographic and Tidal Service"), as well as technical and weather-hydrological reports compiled by territorial agencies and regional databases.

For Liguria, information from "Reports of the weather-hydrological events (2003–2019)" by ARPAL and from "Pluviometrical regional database" by OMIRL was mainly used. For Sardinia, information from "Report of the weather-hydrological events (2013)" by ARPAS was mainly used. For Tuscany, information from "Report of the weather-hydrological events (2009–2019)" by CFR and LaMMA Association was mainly used.

Subsequently, a closer pluviometrical examination has been performed for the most recent and damaging flood events: (i) 4 October 2010 in Sestri Ponente [45], (ii) 18 November 2013 in Olbia [46] and (iii) 9 September 2017 in Livorno [47,48].

Flooded areas in the Sestri Ponente, Olbia and Livorno cities during the three considered events have been surveyed and provided by the regional cartographic online database of Liguria, Sardinia, Tuscany and also by the Livorno Municipality. Culverted streams and canals in the Olbia urban area have been surveyed from aerial photointerpretation, technical surveys and the analysis of technical reports for the construction of the drainage canals.

A multi-temporal comparison has been performed using historical and current topographical and cartographical maps in order to identify anthropogenic landforms and to reconstruct the urban evolution for each case study. In addition, field observations have been carried out to evaluate geomorphological and hydrological aspects of the urbanized areas involved in the flood events. All maps and cartographic data used in the research are listed in Table 3.

We integrated all georeferenced data in a Geographical Information System: using QGIS, we derived new original thematic maps which are useful to provide the immediate identification of both flooded areas in relation to urbanization and the main anthropogenic landforms within each study area.

**Table 3.** Vector and raster data used in the research. Name: DTM, Digital Terrain Model; CORINE, Coordination of information on the environment (European Environment Agency, 1995); TMI, Topographical Map of Italy. Source: GE, Google Earth; IGM, Italian Military Geographical Institute; ISPRA, Higher Institute for Environmental Research and Protection; LR, Liguria Region; PGRA, Flood risk management plan, SR, Sardinia Region; TR, Tuscany Region. Type: R, raster; V, vector.

| Name | Source | Type | Scale/pixel | Date |
|---|---|---|---|---|
| Catchment | LR | V | 1:10,000 | 2019 |
| | LR | R | 5 m | 2016 |
| DTM | TR | R | 10 m | 2017 |
| | SR | R | 10 m | 2012 |
| | LR | V | 1:10,000 | 2015 |
| Flooded areas—PGRA | TR | V | 1:10,000 | 2016 |
| | SR | V | 1:10,000 | 2017 |
| High-resolution satellite image | GE | R | - | 2020 |
| | LR | V | 1:5,000 | 2019 |
| Hydrographical network | TR | V | 1:10,000 | 2018 |
| | SR | R | 1:10,000 | 2019 |
| Land use—CORINE Land cover | ISPRA | V | 25 m | 2012 |
| Soil sealing | ISPRA | R | 10 m | 2018 |
| TMI—Series 25 "Foglio 082—Tavoletta II-NE (Rivarolo Ligure/Sestri Ponente)" | IGM | R | 1:25,000 | 1878, 1907, 1923, 1930, 1934, 1940 |
| TMI—Series 25 "Foglio 082—Tavoletta II-SE (Genova)" | IGM | R | 1:25,000 | 1899, 1907, 1923, 1930, 1934, 1939 |
| TMI—Series 25 "Foglio 111—Tavoletta I-NO (Tombolo/Tirrenia)" | IGM | R | 1:25,000 | 1881, 1939 |
| TMI—Series 25 "Foglio 111—Tavoletta I-NE (Guasticce)" | IGM | R | 1:25,000 | 1881, 1939 |
| TMI—Series 25 "Foglio 111—Tavoletta I-SE (Salviano)" | IGM | R | 1:25,000 | 1881, 1939 |
| TMI—Series 25 "Foglio 111—Tavoletta I-SO (Livorno)" | IGM | R | 1:25,000 | 1881, 1939 |
| TMI—Series 25 "Foglio 182—Tavoletta I-NE (Muddizza Piana)" | IGM | R | 1:25,000 | 1958 |
| TMI—Series 25 "Foglio 182—Tavoletta IV-SO (Loiri)" | IGM | R | 1:25,000 | 1896, 1931, 1958 |
| TMI—Series 25 "Foglio 182—Tavoletta IV-NO (Terranova Pausania/Olbia" | IGM | R | 1:25,000 | 1896, 1931, 1958 |

*3.2. Hydro-Meteorological Data of the Last Flood Events*

Liguria, Sardinia and Tuscany overlook the Ligurian-Tyrrhenian Sea and are three regions that are very prone to violent atmospheric phenomena (see Table 1). Genoa Sestri Ponente, Olbia and Livorno, which are coastal cities that arose at the mouth of rivers, naturally have a long and troubled history of floods that are usually flash floods: their degree of damage has increased year after year in proportion to the degree of urbanization reached by the cities. The areas of river pertinence, that is, those closest to the riverbeds which were once occupied by fields and pastures, have gradually been invaded by buildings; consequently, every time a watercourse overflows during present times, damages result.

This finding stems precisely from the review of the latest cases in the three cities examined. For Genova Sestri Ponente, the last flood of 4 October 2010 was undoubtedly the most serious in terms of damage. The storm cell formed on the first morning of 4 October 2010 stabilized in the neighborhoods of the western city between Pegli, Sestri Ponente and Val Polcevera where it unleashed all its power and caused a true flash flood. In about five hours, between 8 a.m. and 1 p.m., over 400 mm of water fell on the hills behind Sestri Ponente [45]. All the streams reached rapidly exceptional discharges: the

waters violently flooded shops, garages, basements, squares, streets, washed parked cars away and also resulted in a victim.

Olbia has also suffered many flood events in the past: 14 events in the period 1946–2010. However, the severest flood was the last flash flood that occurred on 18 November 2013. Six provinces out of the eight existing on the island were affected: the total damage amounted to about 660 million EUR [46] and 18 were casualties; some were drowned at home and others were dragged by the fury of the streamflows while driving their cars. The Cyclone Cleopatra hit the interior of Sardinia with cumulative rainfall greater than 400 mm: the Olbia rain gauge station recorded a value of 117.6 mm, while the Putzolu rain gauge station, in a village close to Olbia, recorded 175.2 mm in 24 h [47]. The waters of the canals and streams crossing the town overflowed with heights greater than two meters on the countryside level. In the town if Olbia, 11 were the casualties and 40 people were hospitalized for symptoms of asphyxia and hypothermia after having been at the mercy of freezing water for hours. Over 2000 displaced people resulted and some hundreds of millions of EURO in damages were incurred.

Livorno also has a long list of flood events; there is at least 14 between 1946 and 2004, but the most recent event was certainly the most serious of its history. On the evening of 9 September 2017 a violent cloudburst hit the Livorno area: 242 mm of rainfall (74.8 mm in 30 min and 210.2 mm in 2 h) was recorded on the hilly areas just east of the city, on the upper basins of the Rio Maggiore and Rio Ardenza streams [48]. The Rio Maggiore stream, culverted in the 1980s, overflooded at the beginning of the culvert and invaded a large area of the city. In total, over 4.3 km² were flooded and a large part of this was an urbanized area. The damage to structures and infrastructures was very serious (6.6 million euros), with 8 casualties and a dozen injured in the city alone.

## 4. Results

### 4.1. Rainfall Events

With respect to the event of 4 October 2010 on the Ligurian coast, from the analysis of rainfall data recorded in the basins considered (Table 1) it can be highlighted that for the Genoa Sestri Ponente area precipitation resulted from the formation of intense self-regenerating systems (MCS) due to a configuration favorable to a strong convergence between South and South-East, which insisted on the center of the region and in particular on the border between the provinces of Genoa and Savona. Around midnight on 4 October 2010, a stormy event of strong intensity occurred in the area of the Ligurian coast, enhanced by the orographic barrier of the Alpine–Apennine chain and favored by the high sea temperature due to the concomitant presence of an anticyclonic front in the Mediterranean Sea.

After about six hours, a violent weather-pluviometric system reached the town of Varazze with rainfall of about 100 mm/1 h and 220 mm in 3 h. Between 9 p.m. and 12 p.m., the storm cells that had hit the Riviera di Ponente moved towards Genova Sestri Ponente, which was about 20 km to the east. Here, the rainfall recorded at the Mt. Gazzo station (OMIRL-ARPAL hydrological Service) reached 124 mm/1 h, 243 mm/3 h, 360 mm/6 h and 411 mm/12 h (Figures 4–6), compared to annual averages of about 1100 mm. In the areas surrounding Sestri Ponente, high intensities of rain were recorded: 98 mm/1 h peak and a cumulative rainfall of 377 mm/12 h in Pegli (west), while at the Bolzaneto station (north) cumulative rainfall was recorded at 73 mm/1 h and 295 mm/12 h. In Genoa and east of Sestri Ponente, the peak intensity was 40 mm/1 h and the cumulative was 100 mm/12 h (Figure 7).

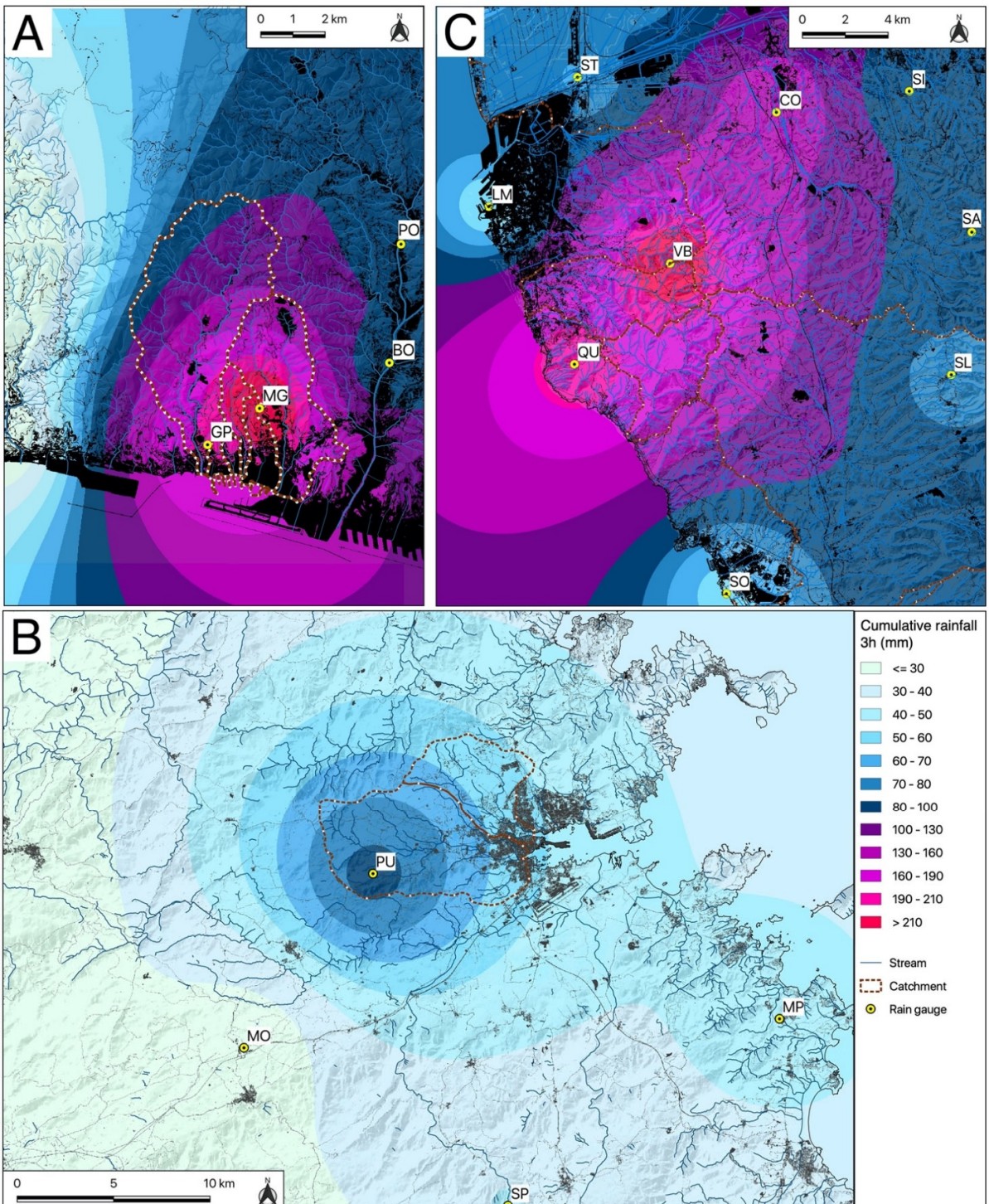

**Figure 4**. Cumulative rainfall map for a time period of 3 h: (**A**) Genova Sestri Ponente, 4 October 2010; (**B**) Olbia, 18 November 2013; (**C**) Livorno, 9 September 2017. Rain gauge stations: (GP) Genova Pegli; (MG) Monte Gazzo; (PO) Pontedecimo; BO) Bolzaneto; (LM) Livorno Mareografo; (ST) Stagno; (QU) Quercianella; (VB) Valle Benedetta; (CO) Collesalvetti; (SI) Siberia; (SA) Santermo; (SL) Santa Luce; (SO) Solvay; (PU) Putzolu; (MO) Monti; (MP) Monte Petrosu; (SP) Sa Pianedda.

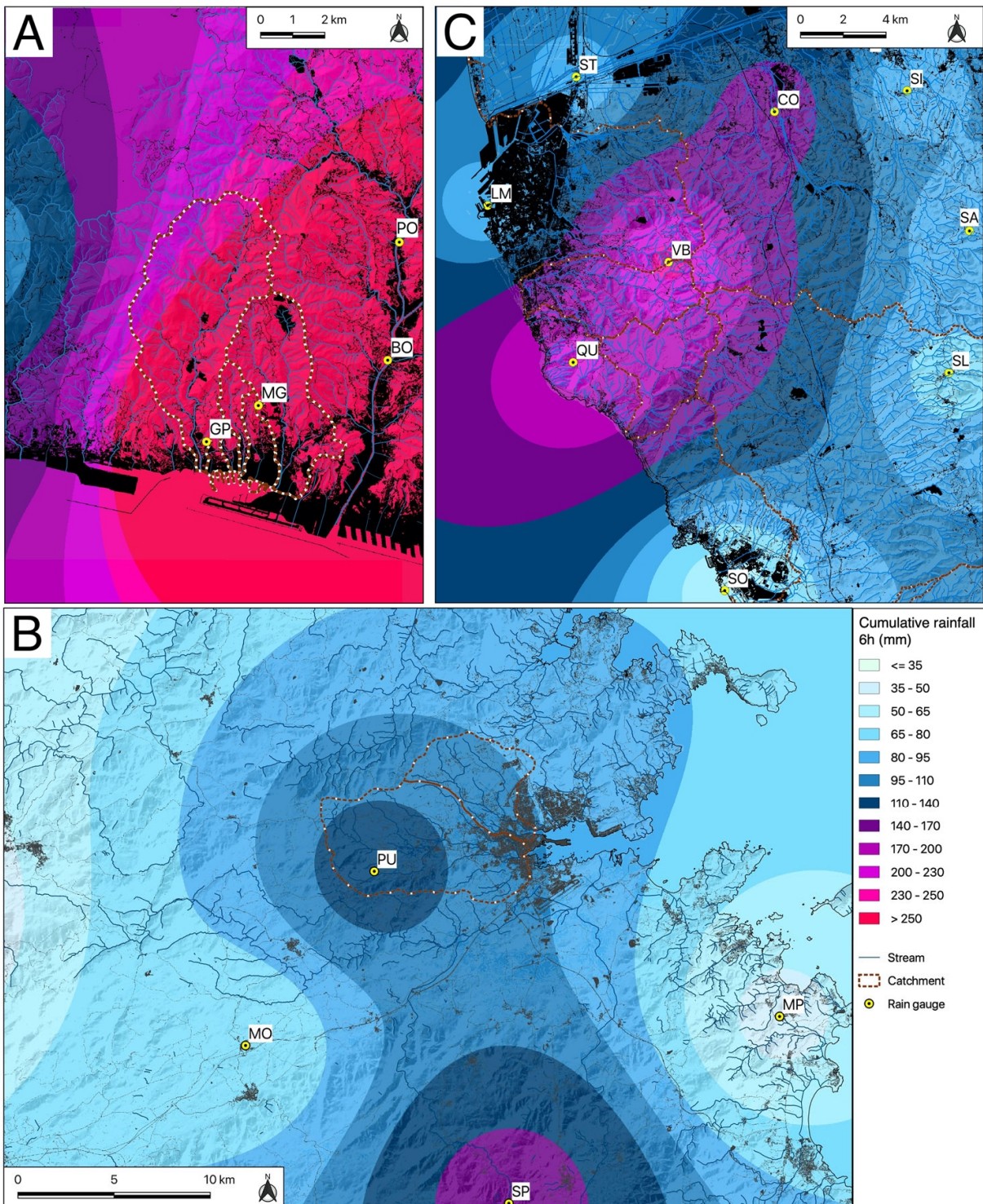

**Figure 5.** Cumulative rainfall map for a time period of 6 h: (**A**) Genova Sestri Ponente, 4 October 2010; (**B**) Olbia, 18 November 2013; (**C**) Livorno, 9 September 2017. Rain gauge stations: (GP) Genova Pegli; (MG) Monte Gazzo; (PO) Pontedecimo; (BO) Bolzaneto; (LM) Livorno Mareografo; (ST) Stagno; (QU) Quercianella; (VB) Valle Benedetta; (CO) Collesalvetti; (SI) Siberia; (SA) Santermo; (SL) Santa Luce; (SO) Solvay; (PU) Putzolu; (MO) Monti; (MP) Monte Petrosu; (SP) Sa Pianedda.

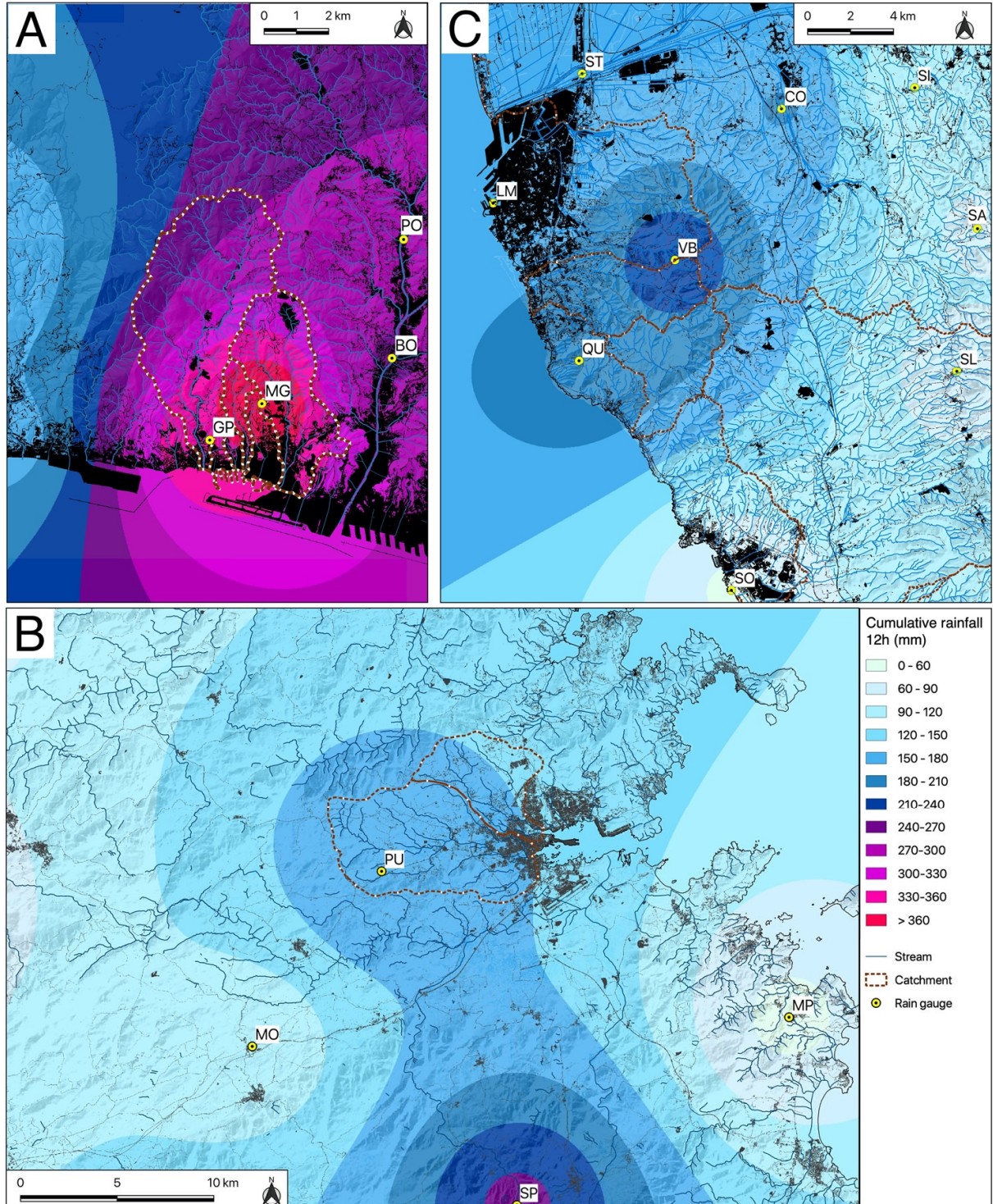

**Figure 6.** Cumulative rainfall map for a time period of 12 h: (**A**) Genova Sestri Ponente, 4 October 2010; (**B**) Olbia, 18 November 2013; (**C**) Livorno, 9 September 2017. Rain gauge stations: (GP) Genova Pegli; (MG) Monte Gazzo; (PO) Pontedecimo; (BO) Bolzaneto; (LM) Livorno Mareografo; (ST) Stagno; (QU) Quercianella; (VB) Valle Benedetta; (CO) Collesalvetti; (SI) Siberia; (SA) Santermo; (SL) Santa Luce; (SO) Solvay; (PU) Putzolu; (MO) Monti; (MP) Monte Petrosu; (SP) Sa Pianedda.

From 1945 to present day, in which at least ten events with damaging effects on the ground occurred, the event of 4 October 2010 in Liguria the sixth highest for rainfall intensity within a time period of 12 h.

The meteorological event that affected the Olbia area on 18 November 2013 is among the most serious in recent years that have extensively affected the entire region of Sardinia in a paroxysmal manner. The set of meteoclimatic conditions resulted in the formation of strong "V-shaped" self-healing marine storms on the eastern sector [28] and is linked to the Cleopatra perturbation with consequent effects on the ground, such as flash floods.

The violent thunderstorms, produced by the convergence line of winds from the west and south-east close to the internal reliefs in the upper part of the river basins, have given rise to cumulative rainfall exceeding 230 mm in the area; an exception to this is Olbia, which hovered around the 120 mm mark with a maximum hourly intensity of 61 mm (Sa Pianedda station) [49]. With respect to the rainfall trend, reference was made above all the rain gauges of Olbia and Putzolu (ARDIS hydrological network): the latter, less than 10 km as the crow flies west of Olbia, is more representative of the hydrological conditions persisting in the interior of the hydrographic basins along which the most damaging effects occurred. The set of data available, however, allowed a better spatial reconstruction of the meteoric event.

At the Olbia rain gauge, the maximum intensity was 28 mm/h (between 6 p.m. and 7 p.m.) and it was preceded by at least another 8 h of precipitation with variable intensity up to 18 mm/h. At the Putzolu rain gauge, higher rainfall heights were recorded, with a daily cumulative of 175.2 mm compared to that of Olbia with 117.6 mm. The maximum hourly intensities in Putzolu reached 45 mm/h between 3 p.m. and 4 p.m., while at the Monte Petrosu rain gauge, the intensity over 15 min was 26.4 mm along the coast. The station that recorded the highest cumulative precipitation was that of Sa Pianedda, which is close to the first hills south of Olbia, with values equal to 150 mm/3 h, 167.2 mm/6 h and 247.4 mm/12 h. The estimated return periods are about 200 years (up to 12 h) for the Putzolu station and about 50 years (up to 6 h on the data) for Olbia [49] (Figures 4–6). Over 80% of the recorded rainfall was concentrated in just over six hours, but persisted for over 12 h (Figure 7); due to their continuity, they were sufficient to cause maximum flows not only in the smallest basins but also in the terminal sections of the largest hydrographic basins.

During the night between 9 and 10 September 2017, the flood event that affected the town of Livorno was anticipated by several storms. During the first, which mainly affected the coastal areas between the territories of Livorno city and Marina di Pisa, maximum cumulative rains of 63.4 mm/1 h was recorded over Livorno (between 8:45 p.m. and 9:45 p.m.) and 65.6 mm/1 h in Marina di Pisa (Bocca d'Arno station). In latter area the rainfall continued to intensify, but the rains practically stopped in Livorno after 9.45 p.m.

Starting from 2:00–2:30 a.m. on Sunday, a new and strong thunderstorm, which then turned out to be the most violent, mainly affected the areas between the southern area of Livorno city and Rosignano town [43,50–52]. In these areas, the values of rainfall reached, which on short durations are really extreme, had peaks higher than 42.4 mm/15′, 121.8 mm/1 h (Rp > 200 y), 210 mm/2 h and 230 mm/3 h [44,50] (Figure 7). There is a clear difference between the maximum data recorded in these hours in the different time intervals of the durations 1, 2 and 3 h by the stations of Quercianella and Valle Benedetta compared to the stations located slightly further south or more inland (such as Castellina Marittima and Santa Luce) (Table 4) or further north (as Livorno Mareografo); this difference highlights the strong localization of the thunderstorm phenomenon that locally discharged over 200 mm of rainfall in 2 h (Figures 4–6). Estimated return periods for 1 h and 3 h of rainfall that were recorded during this event are more than 200 years.

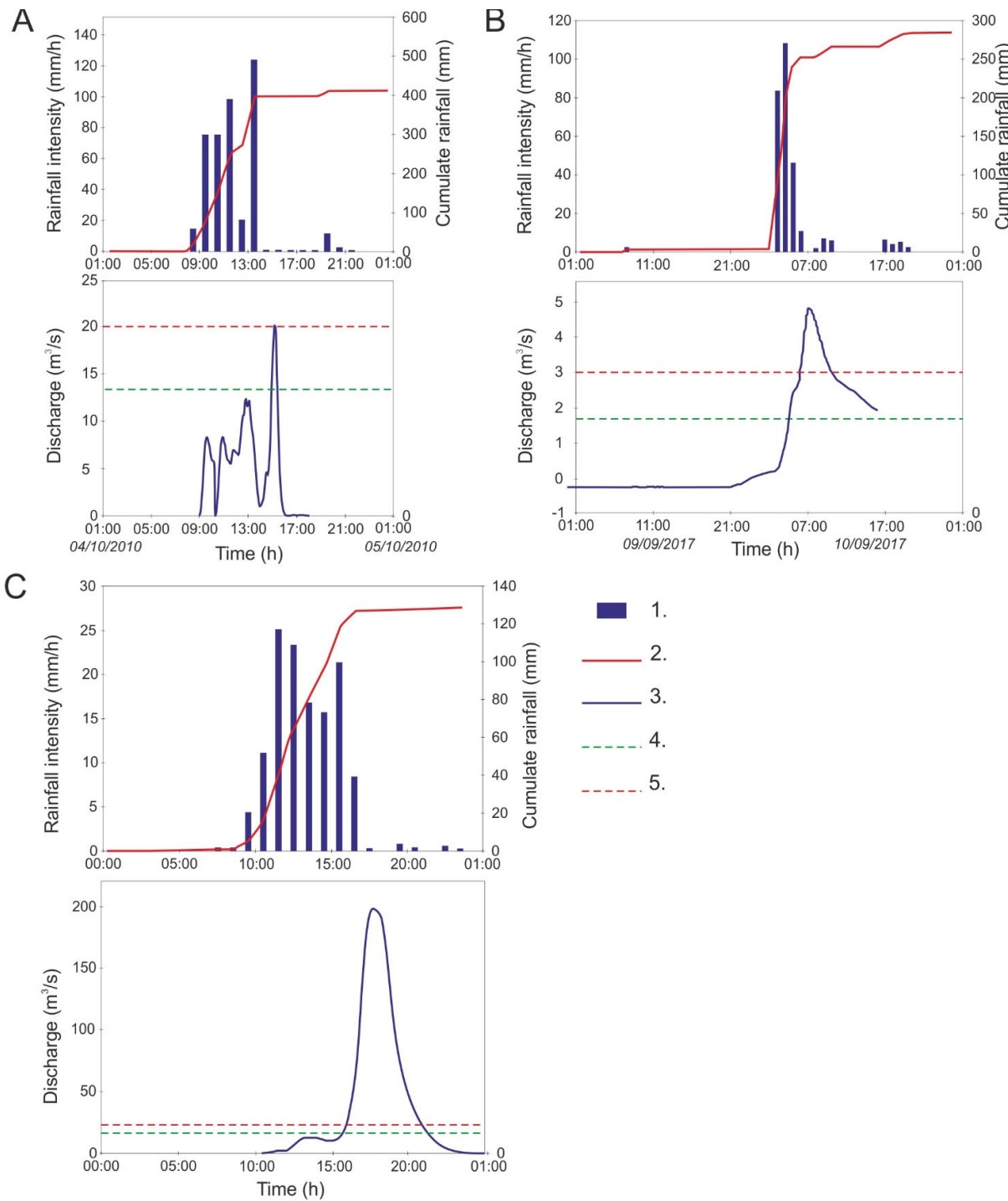

**Figure 7.** Hyetograms and hydrograms for the events of Genoa (**A**), Olbia (**B**) and Livorno (**C**). Legend: 1. Hourly intensity (mm/h), 2. Cumulated rainfall (mm), 3. Discharges (m³/s), 4. Discharges for return period = 50 years (m³/s); 5. Discharges for return period = 200 years (m³/s).

**Table 4.** Maximum cumulative rainfall values over short durations (from 15 min to 3 h) recorded for the stations of Quercianella, Valle Benedetta, Santa Luce and Castellina Marittima [48].

| Rain gauge Station | Max Rainfall 15′ (mm) | Max Rainfall 1 h (mm) | Max Rainfall 2 h (mm) | Max Rainfall 3 h (mm) |
|---|---|---|---|---|
| Quercianella | 42.4 | 121.8 | 188.6 | 206.2 |
| Valle Benedetta | 38.4 | 120.8 | 210.2 | 235 |
| Santa Luce | 23 | 66.4 | 98.2 | 105.8 |
| Castellina Mar. | 40.2 | 89.6 | 109.6 | 122.2 |

*4.2. Ground Effects of the Flash Floods*

With respect to the 2010 event in Genoa, the effects on the ground were determined by rainfalls produced by the convergence between the northern currents coming from the colder Po basin and the warm humid current coming from the sea. In the face of the quantities of rain received from the Sestri Ponente area, the water level in the streams increased in a short time: the Varenna Stream in Pegli reached a peak of 2.62 m with a rapid increase of 2.08 m in one hour (Pegli hydrometer). The Molinassi, Chiaravagna, Ruscarolo and Cantarena streams flooded the areas close to their beds and the Genoese quarter of Sestri Ponente (Figure 8A). Along the streets, the water level reached variable heights from 20 cm to over 150 cm; the estimated discharge rates for the Chiaravagna and Cantarena streams were comparable to those calculated for a Return time (Rt) of 50 years, while for the flood of the Molinassi Stream it was estimated at Rt = 200 years.

The response times of the meteorological-hydrological event were extremely short: Testimonies and amateur images document that after less than half an hour from the peak of flow, the flood of the Chiaravagna stream occurred and it poured along the roads as the water leaking from the streambed. The rainfall caused a rapid increase in flooding along with the transport of suspended materials and floating shrubs and trees that were eroded along unprotected embankments. The speed of the process made it impossible to implement interventions, unless the event was almost concluded. People were taken by surprise along the streets: water spread into residential areas and businesses and caused extensive damage (Figure 8B).

Along the slopes of the Chiaravagna (11 km²) and Molinassi (2 km²) basins, many shallow landslides have been triggered: they interrupted the access roads to the small inhabited areas placed on the hills.

Many streets were flooded and the settlements on the adjacent hills were isolated. The flow of mobilized debris was quickly channeled along the river beds and lower areas which caused critical hydraulic conditions in the secondary hydrographic network and also because the canals that pass culverted under the roads and the inhabited area were not able to dispose the relevant discharges and were quickly clogged.

With respect to the 2013 event, the area affected by the event was estimated at around 1500 km² and includes three main basins: the Cedrino and Posada basins and the catchment upstream of Olbia [53]. The city of Olbia was the most affected city, with eleven victims; much of the downtown area was inundated by the flood waters of the San Nicola and Seligheddu streams in the mouth area. Witnesses claim to have seen the hydrometric levels increase by about 3 m and this would be confirmed by the simulations carried out by [54] and associated with flow velocities higher than 3.2 m/s in the upstream sectors along the hydrographic lines. The railway embankment and the various bridges upon the arrival of the flood wave along the aforementioned canals had a dam effect and caused the flooding of the streets and the first floors of the houses. The most acute phase of the flood event was observed between 5:00 p.m. and 7:00 p.m., with more evident manifestations at around 6:00 p.m. on the Rio Seligheddu Stream and at around 6:30 p.m. on the urbanized stretch of Rio Gadduresu Stream which is its left tributary. Around 9:00 p.m. the flood had subsided with evident manifestations in the sectors surrounding the watercourses of Seligheddu, San Nicola, Zozò, Paule Longa and to a lesser extent the

Pasana. In some sectors (former Artillery area), with variable tie rods up to about 2 m, the effects of the flood of Rio Gadduresu Stream from the north and east overlapped with those of Rio Seligheddu Stream from the south, which is a condition favored by the existing artifacts which represented temporary structures that are damming to the outflow of flood waters (Figure 9A–C) [55].

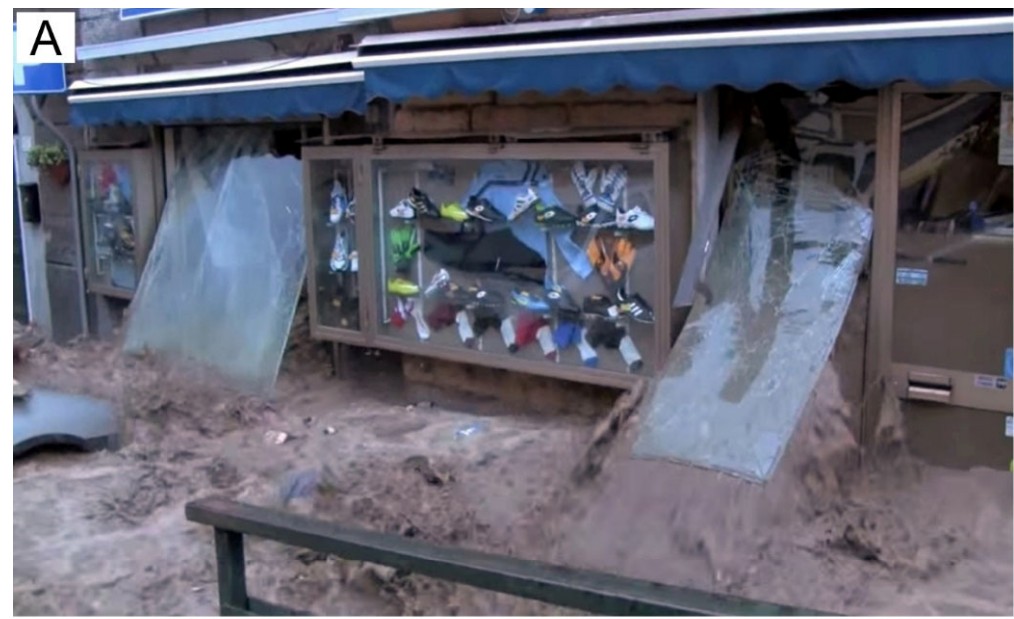

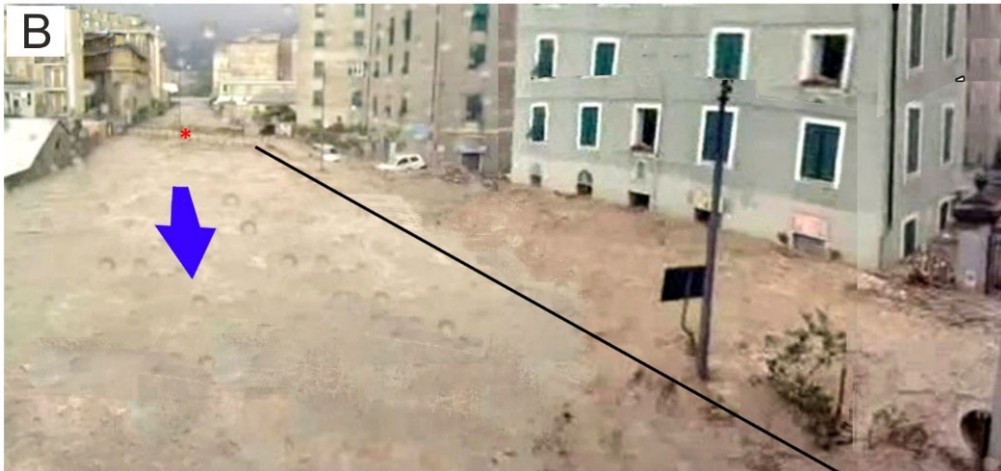

**Figure 8.** (**A**) Chiaravagna Stream flooded the ground floors, where there are many business shops and in some cases the water and mud depending on the preferential outlet flow found that entered the shops from the rear part, as can be seen in a sports shop in the Aprosio Square [56]. (**B**) Image taken from a movie. Chiaravagna Stream during the paroxistic phase of flooding: the black line indicates the submerged left bank wall; the red asterisk represents the bridge of Chiaravagna Street. The building from which the photograph was taken was built just on the riverbed in the 1960s. The October 2010 event (the last one of a long list) was recognized as one of the causes of the flooding of the river: it was demolished a few years later [56]).

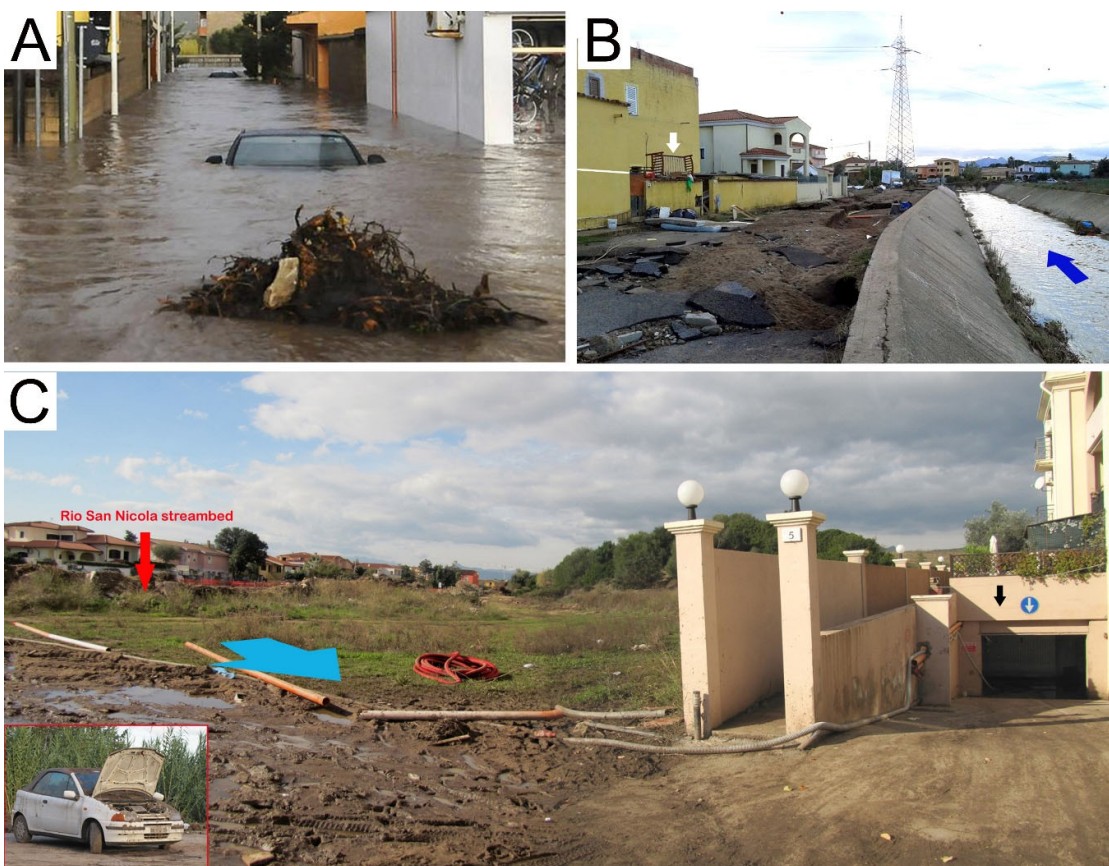

**Figure 9.** (**A**) Olbia: Although about 15 h had passed since the paroxysmal phase, in some morphologically depressed areas of the city the water remained well above 1.3–1.4 m in height. Some cars were totally submerged in one of the streets most affected by the flood in the Baratta quarter (courtesy of private citizen). (**B**) Olbia, left bank of the Rio Seligheddu Stream, canalized stretch (foto Luino). Water level left a marker on the wall, which is evidenced by white line. The height of the waters is evidenced by the bed dragged and placed over the roof of the ground floor (white arrow). (**C**) Olbia: very common situation along the areas close to the streambeds (foto Luino). In this image, the Rio San Nicola (red arrow) waters exceeded the natural bank that was devoid of embankments or retaining walls and easily reached the houses located on the left bank, which was a short distance away. In many houses, the underground garages have been foolishly built and had been totally flooded (black arrow indicates the level), with serious damage inducted to parked cars. The extracted cars are so saturated with fine material (silt and clay) that they had to be demolished (image in the left corner).

During the 2017 event, watercourses flooded the surrounding areas in the hilly sector, where some bridges were damaged and many residents remained isolated. Towards the valley, in the area of the Rio Ardenza and Rio Maggiore mouths, the effects were even more serious with extensive flooding and entire neighborhoods invaded by water and mud. The major effects on the territory (floods, overflows and transport of debris material) were caused by the minor hydrographic network that originates from the Livorno hinterland and flows directly into the sea, as in the case of the Rio Maggiore and Rio Ardenza streams. The culverted stretches, often having insufficient section and occluded by detrital, vegetal and urban material carried by the waters, were bypassed by the floodwaters that retraced the ancient surface river paths. In particular, the waters of the Rio Maggiore Stream, despite the presence of retention basins, managed to flow freely in the area of the Stadio Ardenza District, in the neighboring streets and in Barriera Margherita. The Rio Maggiore Stream escaped from the culvert and poured with high speed into a fenced courtyard that was morphologically depressed compared to the nearby streets. The ground-floor flat in Sauro Street (Figure 10A,B) was flooded within minutes and four people drowned in it. Due to the overflow of the Ardenza Stream and

its tributary Forcone stream, four other people lost their lives. The flood event caused in total eight victims in the Livorno area.

In addition to the Rio Ardenza and Rio Maggiore streams, the Ugione, Quercianella and the Chioma streams inundated large areas (Figure 10C). A bridge adjacent to a provincial road collapsed along the Ardenza. The total damage was estimated at 180 million EUR [57].

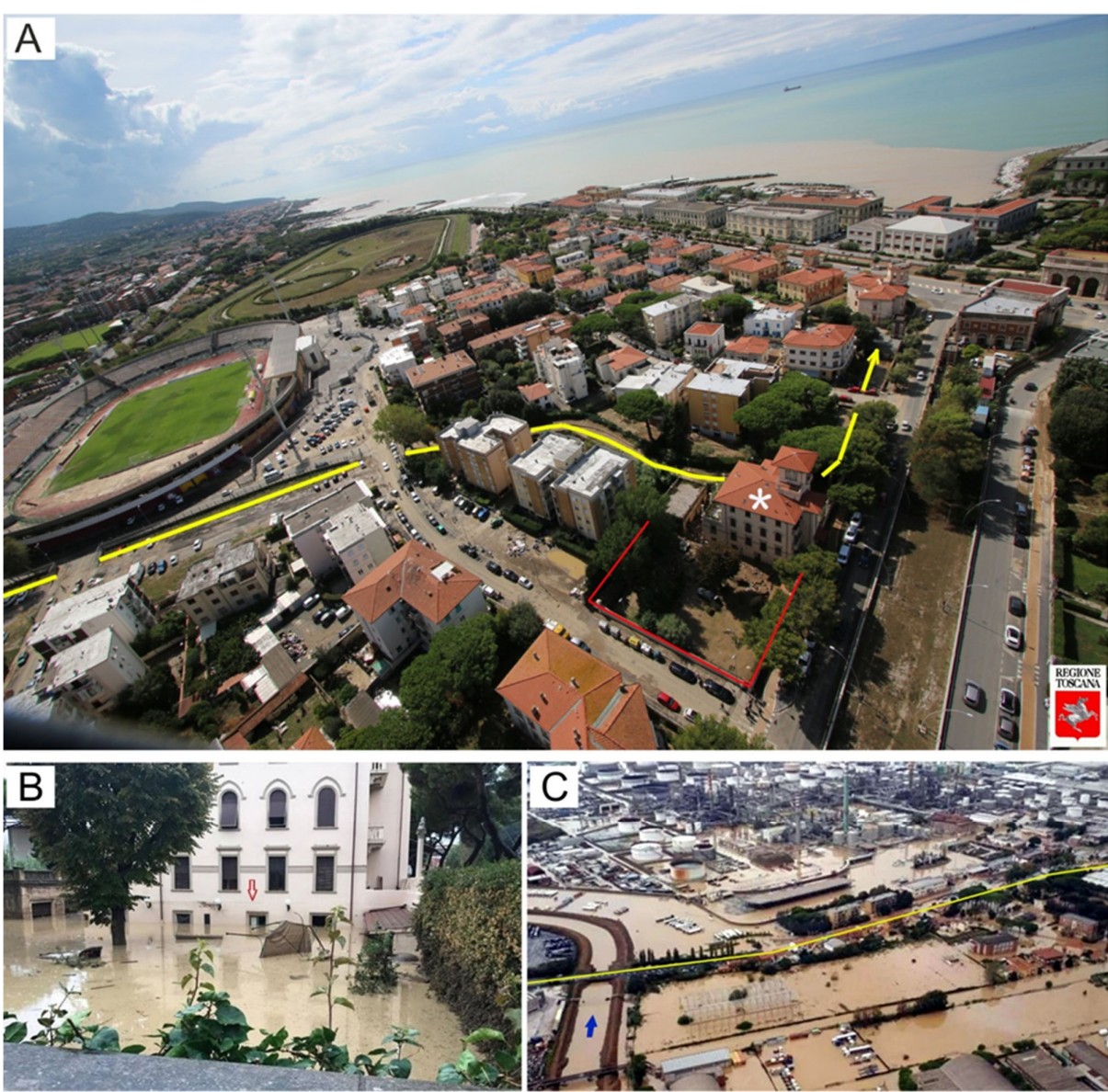

**Figure 10.** Livorno. (**A**) aerial photograph of the Ardenza Stadium district. In the foreground is the house where the four victims drowned (white asterisk). The yellow line highlights the culverted streambed of the Rio Maggiore: its flood waters were the cause of the rapid flooding of the courtyard (evidenced by red lines) and of the ground-floor flat [58]. (**B**) Livorno: (a) the house where four people from the same family drowned (courtesy of Il Tirreno). Their ground-floor flat is located at the end of a large courtyard, below street level on Rodocanacchi Street (area of the Ardenza football stadium). The red arrow indicates the level reached by the floodwaters in a few minutes. (**C**) Livorno: the Stagno district that was largely flooded by the Rio Ugione floodwaters with the Via Aurelia (yellow line) and the refineries in the background [58].

### 4.3. Urban Geomorphology

The case studies examined are characterized by a similar geomorphological structure and recent evolution, which has profoundly changed almost all Mediterranean urban areas [22,29]. Sestri Ponente, Olbia and Livorno are in fact three cities built on a coastal floodplain at the mouth of small hydrographic basins (Figure 11A–C).

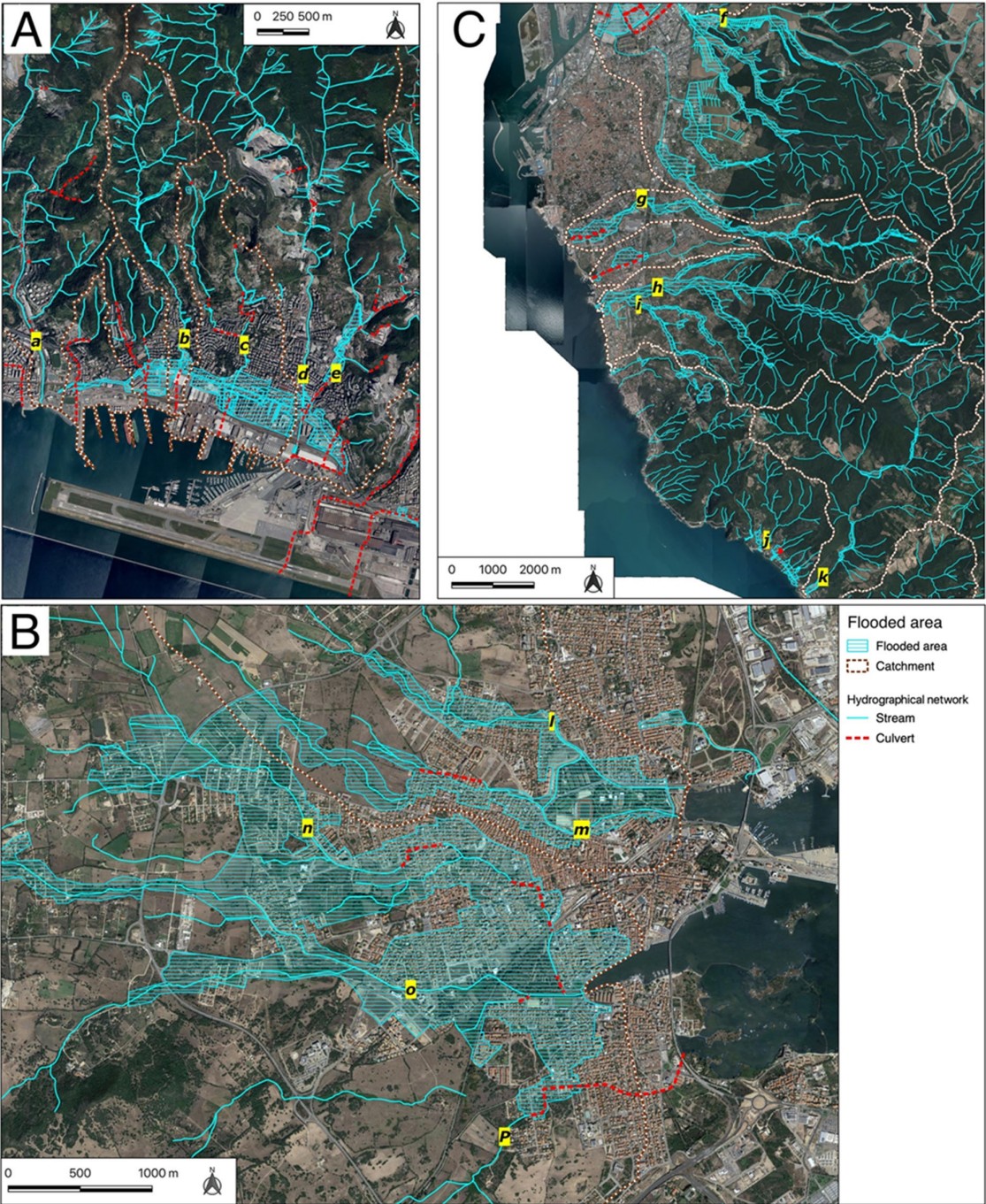

**Figure 11.** Hydrographic networks: culverted stretches are indicated with red lines. (**A**) Genoa Sestri Ponente; (**B**) Olbia; (**C**) Livorno. Streams: (a) Varenna; (b) Molinassi; (c) Cantarena; (d) Chiaravagna; (e) Ruscarolo; (f) Ugione; (g) Maggiore; (h) Ardenza; (i) Forcone; (j) Quercianella; (k) Chioma; (l) San Nicola; (m) Canale Zazà; (n) Gadduresu; (o) Seligheddu; (p) Paule Longa.

In this morphological situation, it is therefore possible to identify the processes related to the shapes of the landscape: the sea wave action along the coastal strip and the river dynamics in the rear belt, up to the hill base. The footprint of the anthropic landforms dominates the current urban landscape. If natural landforms were almost exclusive until the 18th–19th centuries, then in the last 150 years there has been a gradual increase in anthropogenic activities in the area with consequent major changes [59]. Two historical moments demonstrate great changes in the urban landscape due to anthropic impact: the first around the middle of the nineteenth century (industrial revolution) and a second in the second half of the twentieth century [60], after the conclusion of the Second World War (greater well-being, population growth and finally tourism). At present time, in all the cases examined it is therefore possible to identify modified natural landforms, anthropic landforms and disappearing (or vanished) landforms. The most common modified natural form is the main and secondary riverbed: in all the cases analyzed, the riverbed in the coastal floodplains was narrowed, channeled, rectified, often culverted and sometimes diverted.

In Sestri Ponente, the Chiaravagna Stream is channeled in its 2 km terminal, while the Ruscarolo Stream, originally an autonomous basin, flows entirely into the Chiaravagna. The other four streams of the plain are invisible, that is, culverted practically throughout the terminal stretch and flow within the urban fabric. A multitemporal cartographic comparison allows us to evaluate a narrowing of the riverbed in the last 200 years between 25 % and 50% [34].

In Olbia, the Seligheddu Stream to the south and the San Nicola Stream to the north are represented today by artificial canals cemented with an inverted trapezoidal section, with many deviated and culverted sections, and affects the town with a multi-kilometric development [49].

The coastal stretch between Livorno and Antignano is characterized by three watercourses: to the south, the Rio Ardenza Stream is channeled in its final stretch for at least 2 km, with conspicuous narrowing and rectification of the section of the riverbed; on the other hand, the Rio Maggiore Stream and the Fosso Botro canal have been transformed into culverts [43].

The anthropogenic shapes dominate the urban landscape and can be traced back to forms of accumulation: in Sestri Ponente, Olbia and Livorno, among the most significant and common are the embankments, the reclaimed land at sea and the defenses along the coast, while urbanization has involved remodeling of the existing topographic surface with diffuse fillings.

Sestri Ponente is dominated by the sea fill on which the shipyards and the Genoese airport are built on: the railway line runs on an embankment and, further upstream, there is the motorway embankment. The fluvio-coastal plain is practically fully urbanized and the most significant phases of expansion were those of the second half of the 19th century and in the second half of the 20th century.

Olbia has a deeply modified coastal strip for the construction of the maritime station by the filling of the sea; the road and railway embankments emerge in the strip behind the historic core. Particularly significant is the construction of artificial canals built in the 1920s, while urbanization appears significant from the second half of the 20th century, but continued into the third millennium.

The urban coastal stretch south of Livorno has sea fills that are almost continuous, but they are less deep than the ones in Sestri Ponente and are protected by sea defenses. In the strip immediately behind it, there are the railway and motorway embankments; urbanization is continuous, with the exception of some sports facilities, and can be traced back to the second half of the 20th century. The waterways have been channeled and retention basins have been built along the final stretch of the Maggiore Stream.

Lastly, the disappeared forms, i.e., those dismantled or covered by anthropic activities, deserve a mention: In the case of Sestri Ponente, in addition to the consumption of land on the entire coastal floodplain there is the disappearance of the beach that

occupied the entire stretch of sea in proximity to the coast for a length of about 2.5 km. In the Livorno coastal strip, the small cliffs modeled in the cemented sandstones alternating with small beaches have been incorporated into the sea fillings built from the second half of the 19th century, while in the coastal plain behind them marshes and coastal dunes have disappeared. In the case of Olbia, even if the imprint of the rias coast remains evident, the disappearance of the marshes has been noted, which characterized much of the coastal strip north and south of Olbia (Terranova Pausania) and into which the hydrographic network used to flow, as well as the salt flats near the historic center and small beaches bordered by cliffs modeled in granite between the Roman Port in the north and the Lepre Island in the south.

## 5. Discussion

The case-studies presented in this article show some geomorphological similarities: the cities arose on alluvial plains, typical of the coastal strip [22,29,56,61] of the Ligurian-Tyrrhenian Sea. They extend for several kilometers along the coastline (2.6–11.5), occupying variable areas (8–30 km²) with high hills behind (330–580 m) and some kilometers away (3.8–6). Many streams cross the cities and some of them possess relevant areas (up to 38 km²): they can reach relevant discharges during the violent rainfall events and in proportion to the basin areas (up to 8.6 m³/sec/km²) (Table 5).

With respect to the weather-hydrological aspect, it should be emphasized that all the considered events occurred during the autumn season (from September to November), in conditions of a storm system triggered by cyclogenesis at the Gulf of Genoa (Liguria and Tuscany) or by the extra-tropical cyclone Cleopatra (Sardinia).

**Table 5.** Geomorphological features of the three cases analyzed. City Develop, development of the city along the coastline; Hill Height, height of the hills located behind the city; Distance, distance between the coastline and closest hills; Basin Area, area of the largest hydrographic basin behind the city; Max Discharge, maximum discharge for a 200-years return period.

| City | City Area (km²) | City Develop (km) | Hill Height (m) | Distance (km) | Basin Area (km²) | Max Discharge (m³/sec) |
|---|---|---|---|---|---|---|
| Genova Sestri Ponente | 8 | 2.6 | 580 | 3.8 | 11 (Chiaravagna) | 213 (50 yrs) |
| Olbia | 27 | 11.5 | 460 | 6 | 38.4 (Seligheddu) | 330 |
| Livorno | 30.2 | 7.8 | 330 | 6 | 33.2 (Ugione) | 137 |

Despite the trend in the number of rainy days being negative and the progressively decreasing annual cumulative rainfall , perturbations capable of generating intense rainfall are increasingly frequent in the Mediterranean area [10], with a growth corresponding to intense geo-hydrological events. A sentence that summarizes this concept is currently widely used: "it rains less, but worse".

If we add the progressive increase in temperatures to this [62], it is possible to confirm the data on climate change underway [63,64] with recent evidence of events studied in the geo-hydrological field: For example, the event of Lavagna-Genoa in 2002 [30], autumn 2011 in Liguria [65] and event of October 2014 in Liguria [33,66]. These data are also supported by studies in other disciplines [67,68].

The events considered had rainfall that reached values between 10.4–15.2% of the annual total in 1 h; between 22% and 25.8% in 3h (minimum difference between the minimum and the maximum); between 26.5% and 32.7% in 6 h; finally, values between 27.2% and 42% in 12 h (Table 6).

The percentages obtained actually indicate the rainfall characteristics of each individual event: the Livorno event was more concentrated on 1–3 h, the Sestri Ponente on 6 h and the Olbia event on 12 h. The intense precipitations occurred in correspondence with strong winds and storm surges, showing hourly cumulative rainfall values comparable to half-yearly or annual averages. The precipitations sent the hydrographic system into crisis: the levels of the streams grew rapidly, reaching and exceeding the alert levels.

**Table 6.** Rainfall of the three events (for 1,3,6 and 12 h) compared with the mean annual precipitation, MAP (%).

|  | MAP | 1 h | 3 h | 6 h | 12 h |
|---|---|---|---|---|---|
| Genoa Sestri Ponente | 1100 | 124 (11.3%) | 243 (22%) | 360 (32.7%) | 411 (37.4%) |
| Olbia-Sa Pianedda | 588 | 61 (10.4%) | 150 (25.5%) | 167.2 (28.4%) | 247.4 (42%) |
| Livorno-Quercianella | 800 | 121.8 (15.2%) | 206.2 (25.8%) | 212.4 (26.5%) | 217.6 (27.2%) |

The responses of river basins to intense and short rainfalls depended on several factors: (i) land use, (ii) bedrock permeability, (iii) thickness of the eluvium-colluvial cover and iv) initial content of soil moisture. If, in normal conditions, the Mediterranean catchment areas possess a runoff coefficient of 0.4–0.6, in the conditions of saturated or impermeable soils, runoff coefficients close to 1 can be achieved [69,70].

The streambeds in the cities, over the years, have been gradually narrowed to conquer more urban areas and often the beds have been culverted for long stretches and flow under roads and buildings. It is therefore natural that the streambeds were not able to dispose of the huge discharge of the streams in which water shrubs, large trees (uprooted upstream or in the riverbed), garbage cans and vehicles are often transported.

The increase in urbanized areas has amplified this problem, leading to an increase in waterproofed surfaces; this has caused the irreversible loss of soil and consequent impact on the flow of water, which when unable to infiltrate the soil, is dispersed by surface flow. The latter, increasing in terms of drained volumes and transit speed, is responsible for problems in the control of surface waters, in particular during particularly intense rainfall phenomena. The growth in waterproofed surfaces, in fact, involves an increase in the runoff coefficients and a reduction in the run-off times, making it necessary to construct structures for containment and disposal (bypasses) of exceptional flood events.

In the last 150 years, the Sestri Ponente, Olbia and Livorno cities have suffered remarkable transformations, especially in the coastal areas (Figure 12). This has led to the growth of urbanized areas along the plain, with partial or total impairment of the areas of fluvial pertinence. The waterways have undergone riverbed narrowing, containment and lateral and bottom constraints, which has resulted in the unsuitability for the disposal of significant flood flows.

The serious damage recorded for the cases treated, primarily the 27 overall casualties of the three events described, are linked to the vulnerability of the numerous elements present in the fluvio-coastal plains of Genoa, Olbia and Livorno. Among the anthropogenic forcings that have influenced the hydro-geomorphological dynamics and which have determined the increase in risk conditions, the following are counted [71,72] (Figures 13 and 14):

- Modification of land use in general from agricultural to compact urban cover, with a consequent high decrease in the run-off time;
- Prograzione of the coastline by the method of sea filling (e.g., sea filling for the construction of Cristoforo Colombo platform airport and the marine port of Sestri Ponente);
- Modified river forms: the watercourse active beds show marked evidences of canalization and narrowing of the outflow sections, sometimes deviations and often culverts (Figure 15);

- Construction of buildings and infrastructures (especially since the 1960s) in areas known to be hazardous from a hydraulic point of view.

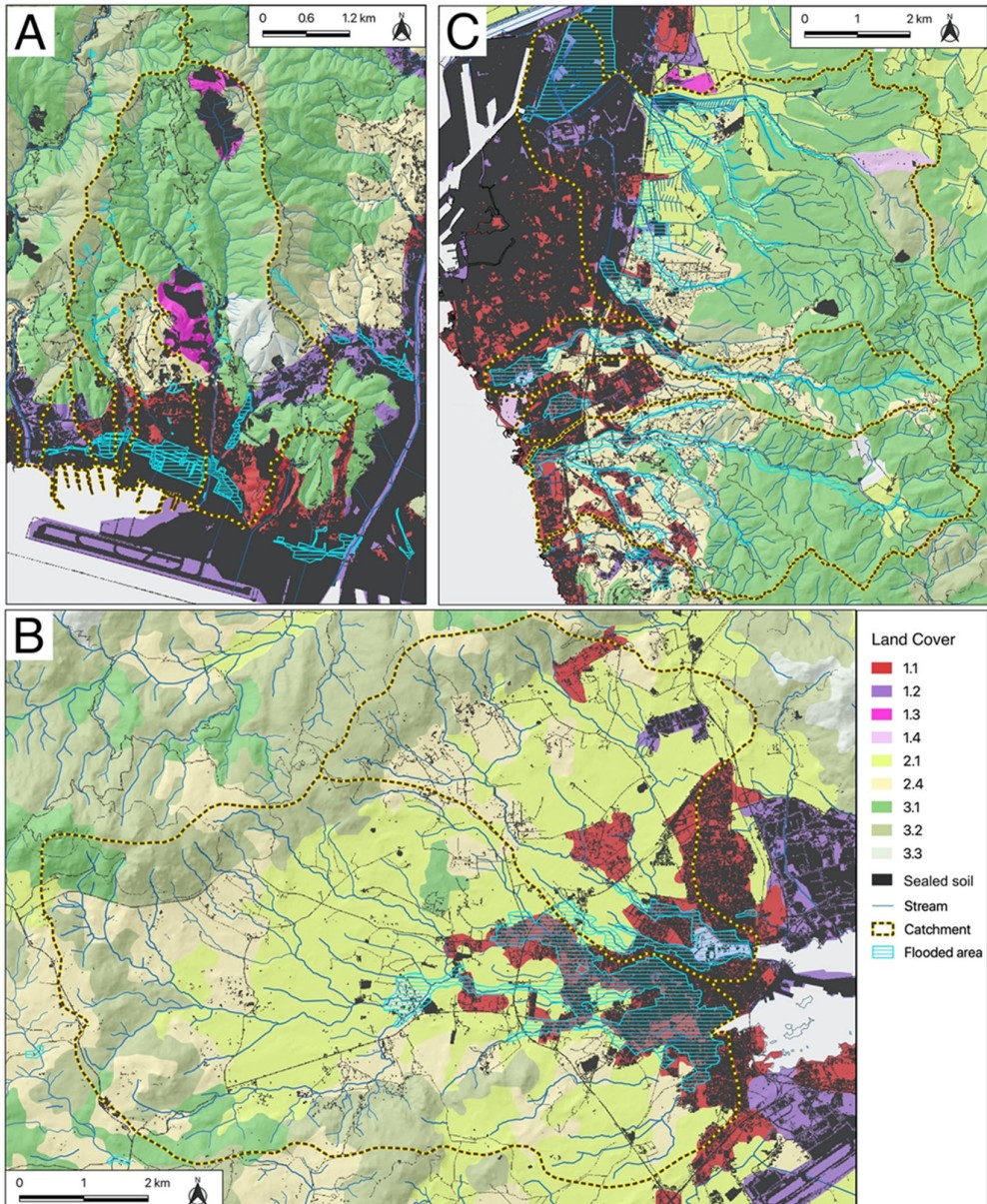

**Figure 12.** Hydrographic networks, urbanized areas and land use of the basins of Genoa Sestri Ponente (**A**), Olbia (**B**) and Livorno (**C**). Land cover: (1.1) urban fabric; (1.2) industrial; (1.3) mine, dump and construction sites; (1.4) artificial non-agricultural vegetated areas; (2.1) arable land; (2.4) agricultural areas; (3.1) forests; (3.2) shrub and/or herbaceous vegetation associations; (3.3) open spaces with little or no vegetation.

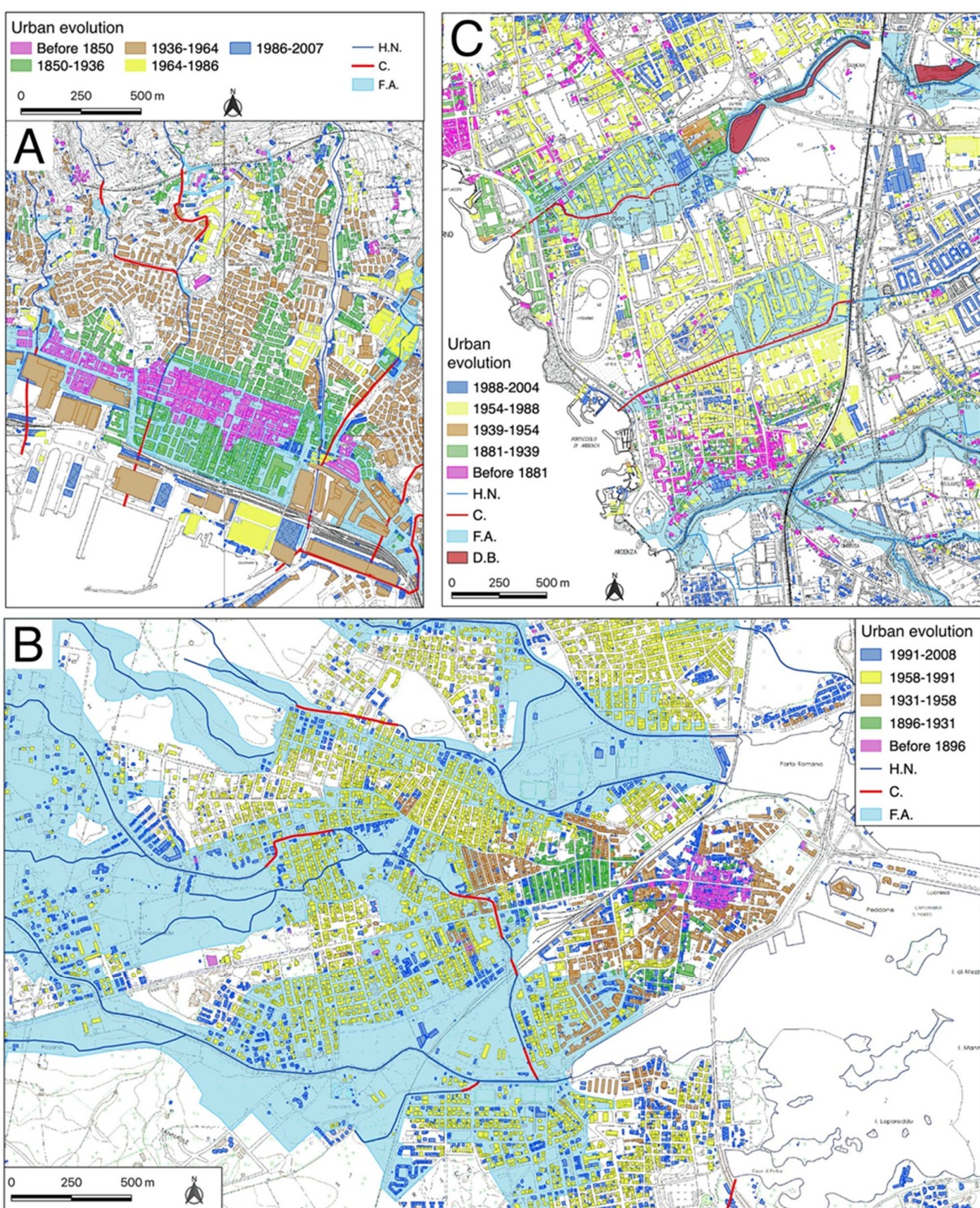

**Figure 13.** Urban evolution, evidenced by different colors for Genoa Sestri Ponente (**A**), Olbia (**B**) and Livorno (**C**) since the end of the 19th century. In the legend: (H.N.) Hydrographical network; (C) Culvert; (F.A.) Flooded area.

A very important aspect that characterized all three events was the lack of communication with the population before and during the culminating phase of the event [73–75]. In all three cities, the inhabitants were surprised by the flooding of the streams

and the rapid growth of water in the city: this made it impossible, in some cases, to escape to safety (15 drowned victims) or to bring their own goods (especially vehicles).

A last and very important topic concerns the historical sources. Moreover, in this study, a careful historical reconstruction has made it possible to detect how much the anthropic changes may have influenced the bed of the watercourses year after year, decisively conditioning the floods during the paroxysmal phase of the event and creating casualties and a lot of damage. A correct utilization of the historical sources could save lives and goods.

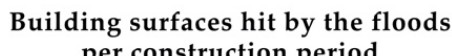

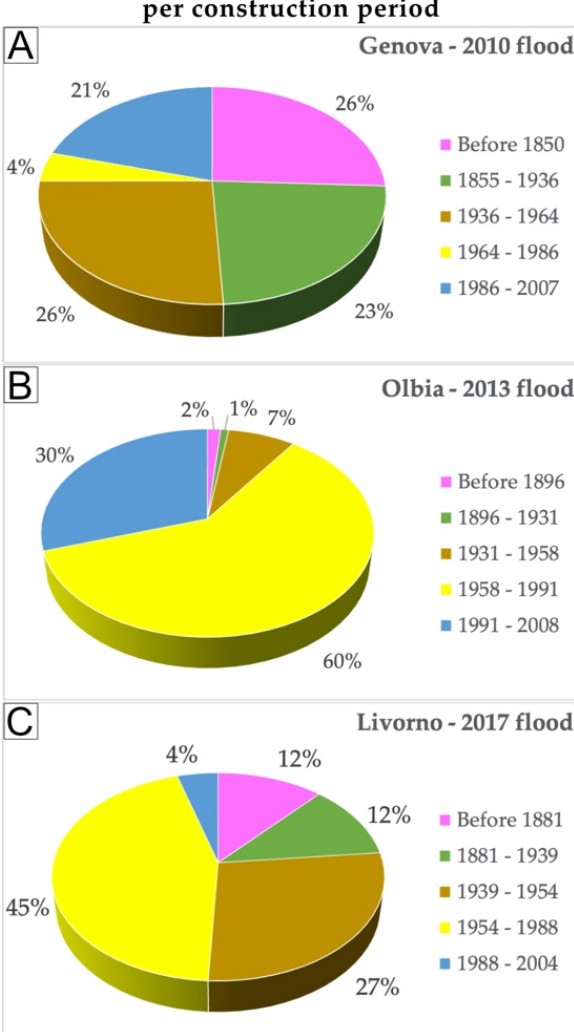

**Figure 14.** Urban development in the coastal plain of: (**A**) Genoa Sestri Ponente, compared with the percentage of urbanization involved over time by a flood of comparable size in terms of volume and area to that occurred 04/10/2010; (**B**) Olbia, compared with the percentage of urban areas involved over time by a flood of a comparable extent in terms of volume and area to that of 11/18/2013; (**C**) Livorno, compared with the percentage of urbanization involved over time by a flood of comparable magnitude in terms of volume and area to that of 09/10/2017.

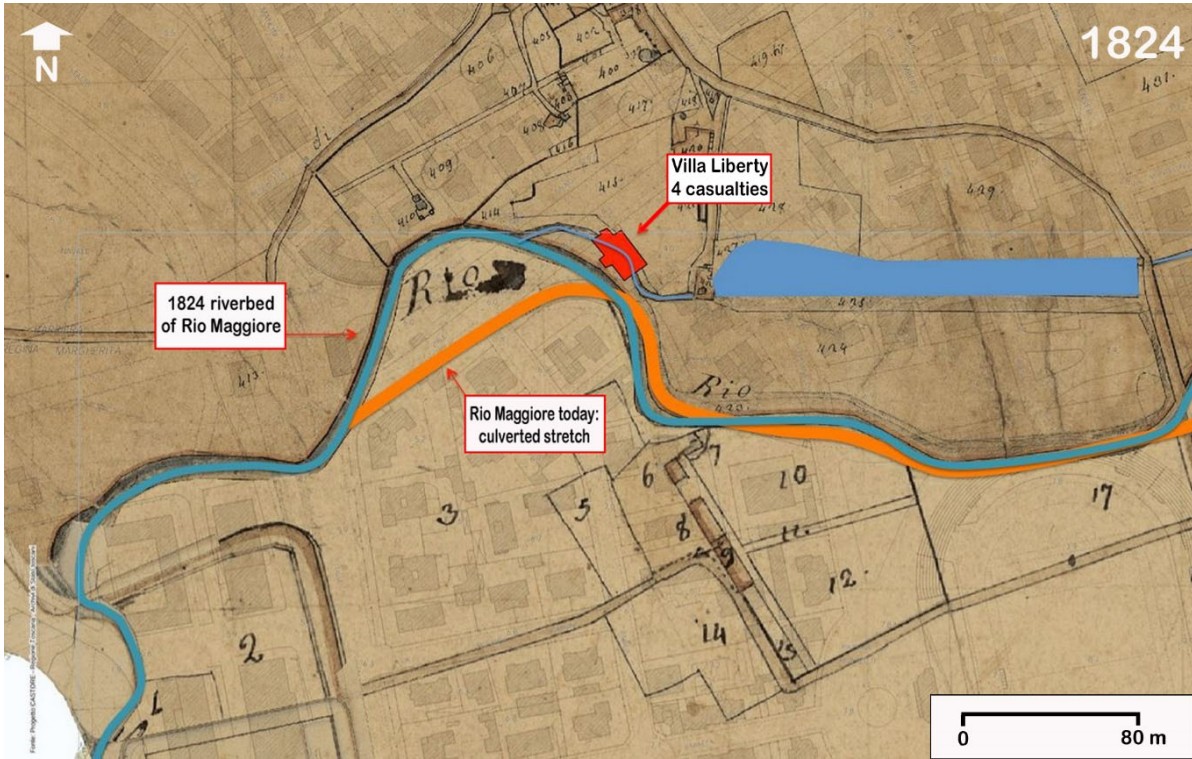

**Figure 15.** The importance of the historical sources is underlined by this 1824 cadastral map of Livorno. The Rio Maggiore Stream flowed naturally and was free from anthropogenic influences (blue stroke on the map). Its course has been diverted over the years and, above all, the stream has been culverted: it now flows invisibly between the houses of the city (orange stroke). The house where the victims drowned in September 2017, which will be built afterwards in the early 20th century, is highlighted in red ([76], modified).

## 6. Conclusions

Two factors, which are very different from each other, are at the basis of the geo-hydrological processes that have affected long stretches of the Mediterranean coast in recent decades. The first, which is of anthropogenic origin, is the urban expansion that began after the end of the Second World War. The second, which is of "apparently" natural origin, is climate change (for which the role played by man is now very clear to everyone). The combination of these two factors, which are in some manner very distant from each other, is now causing a situation that is difficult to manage and that places us at the forefront of real emergencies for which it will be necessary to utilize precise analytical tools, sensible land use planning and rapid and resolving interventions.

The floods that occurred in Genoa, Olbia and Livorno revealed the state of vulnerability of the city towards hydraulic processes in spite of many accurate mitigation interventions carried out in the last decades and above all the inadequacy of the current urban fabric in relation to the hydrographic network, both on the hydraulic and geomorphological level.

The comparison between the events described highlights the need to plan flood risk mitigation activities beyond the essential structural interventions on the hydrographic network, whether it be main and minor and at basin scale (extraordinary maintenance), which involves inevitable financial programs that are costly and delayed over time. It appears essential that we prepare non-structural measures which includes both active (routine maintenance, thickening of the weather-hydrological monitoring and construction of expansion tanks) and passive (land use rules with intensive use of urban drainage systems sustainability, foreclosure of fluvial pertinence areas, historical

investigation of past events, population trainings, information to inhabitants and insurance coverage) measures.

With regard to the actions to be taken in the short-term and medium-term in order to adapt to the resurgence of flash flood effects on the ground, there are several methodological examples for the Mediterranean area that involve urban, social and economic choices. For example, we want to mention the TRIGEAU project [77] and the ADAPT project [78], which suggests strategies to mitigate the effects of climate change. In particular, the ADAPT project suggests: (1) Actions to improve geo-hydrological conditions, such as increasing knowledge on the processes that contribute to the occurrence of geo-hydrological criticalities, adapting existing mitigation works, carrying out interventions for urban flood mitigation and urban greening interventions; (2) actions to increase the resilience of the population and assets at risk with training activities; (3) actions to improve governance with legislative adjustments, urban interventions and limitations on urbanization and restoration of areas of river pertinence and re-naturalization.

Until the state and every single municipality decides to take serious and decisive action to solve this age-old problem, the flash floods will continue to cause victims and massive damage to private and public property.

**Supplementary Materials:** The following are available online at www.mdpi.com/2073-445X/10/6/620/s1, Table S1: Land use in the studied catchments, Table S2: Dates of post-World War II flood events in the three studied cities.

**Author Contributions:** Conceptualization, F.L., F.F. and L.T.; methodology, F.L., F.F. and L.T.; software A.R. and G.P.; validation, A.R., G.P., F.L., F.F. and L.T.; investigation, G.P., F.L., F.F. and L.T.; resources, A.R., G.P., F.L., F.F. and L.T.; data curation, A.R., G.P., F.L., F.F. and L.T.; writing—original draft preparation, F.L., F.F. and L.T.; writing—review and editing, F.L., F.F. and L.T.; visualization, A.R., G.P., F.L. and L.T.; supervision, G.P. and L.T.; project administration, F.L. and L.T. All authors have read and agreed to the published version of the manuscript.

**Funding:** This research received no external funding.

**Institutional Review Board Statement:** Not applicable.

**Informed Consent Statement:** Not applicable.

**Data Availability Statement:** Not applicable.

**Conflicts of Interest:** The authors declare no conflicts of interest.

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
