# Peer review of "Flash Flood Events along the West Mediterranean Coasts: Inundations of Urbanized Areas Conditioned by Anthropic Impacts"

_land, doi:10.3390/land10060620_

Round 1

Reviewer 1 Report

The reviewed paper presents interesting results of an in-depth case study stressing the outstanding flash flood events in the West Mediterranean coasts. The paper is based on a really good study, and it is informative. Its relevance to Land journal is evident. Nonetheless, there are some issues for improvement before further consideration of this paper. Please, find my recommendations below.

First, Figure 12 must be improved: I advise putting the titles outside, putting the same font as the whole article, putting the legend at the bottom and optimizing the space. I think using a full page for that figure is too much.

Second, although the article focuses on the West Mediterranean coast, given that it is an international journal, it would be interesting for the authors to mention analogous European areas in terms of flash floods and urbanization processes associated with tourism, similars the case of Olbia. For example the Canary Islands: [López Díez, A.; Máyer Suárez, P.; Díaz Pacheco, J.; Dorta Antequera, P. Rainfall and Flooding in Coastal Tourist Areas of the Canary Islands (Spain). Atmosphere 2019, 10, 809. https://doi.org/10.3390/atmos10120809] and [Génova, M., Máyer, P., Ballesteros-Cánovas, J., Rubiales, J. M., Saz, M. A., & Díez-Herrero, A. (2015). Multidisciplinary study of flash floods in the Caldera de Taburiente National Park (Canary Islands, Spain). Catena, 131, 22-34.] Or the case of Madeira in Portugal [Fragoso, M., Trigo, R. M., Pinto, J. G., Lopes, S., Lopes, A., Ulbrich, S., & Magro, C. (2012). The 20 February 2010 Madeira flash-floods: synoptic analysis and extreme rainfall assessment. Natural Hazards and Earth System Sciences, 12(3), 715-730.]

I think this work is very interesting, congratulations to all the authors!

Author Response

The reviewed paper presents interesting results of an in-depth case study stressing the outstanding flash flood events in the West Mediterranean coasts. The paper is based on a really good study, and it is informative. Its relevance to Land journal is evident. Nonetheless, there are some issues for improvement before further consideration of this paper. Please, find my recommendations below.

First, Figure 12 must be improved: I advise putting the titles outside, putting the same font as the whole article, putting the legend at the bottom and optimizing the space. I think using a full page for that figure is too much.

R: Thanks for this suggestion. We have changed figures according the advice.

Second, although the article focuses on the West Mediterranean coast, given that it is an international journal, it would be interesting for the authors to mention analogous European areas in terms of flash floods and urbanization processes associated with tourism, similars the case of Olbia. For example the Canary Islands:

[López Díez, A.; Máyer Suárez, P.; Díaz Pacheco, J.; Dorta Antequera, P. Rainfall and Flooding in Coastal Tourist Areas of the Canary Islands (Spain). Atmosphere 2019, 10, 809. https://doi.org/10.3390/atmos10120809] and

[Génova, M., Máyer, P., Ballesteros-Cánovas, J., Rubiales, J. M., Saz, M. A., & Díez-Herrero, A. (2015). Multidisciplinary study of flash floods in the Caldera de Taburiente National Park (Canary Islands, Spain). Catena, 131, 22-34.]

Or the case of Madeira in Portugal

[Fragoso, M., Trigo, R. M., Pinto, J. G., Lopes, S., Lopes, A., Ulbrich, S., & Magro, C. (2012). The 20 February 2010 Madeira flash-floods: synoptic analysis and extreme rainfall assessment. Natural Hazards and Earth System Sciences, 12(3), 715-730.]

R: OK, thanks. We have added these interesting papers.

Reviewer 2 Report

At the first I would like to thank Authors for hard work and very interesting manuscript. 
I have read the paper carefully. The proposed topic is very important in hydrology because the urban flooding is still unsolved problem. In my opinion the presented article could be very interesting for researcher, who deal the similar problems. Nevertheless I have some comments that I would like the Authors respond to.
1.    In my opinion the Introduction must be re-written. Generally the Authors should be focused on flash floods. Currently there is too much general information about floods. The Authors must provide information about urban flooding (what are the main physiographic and meteorological factors causing them, how we can, how we can defend against them – rainfall water management, e.g.). 
2.    Pages 3 and 4: in my opinion the table 1 is not necessary. It should be removed. The same figure 1. Maybe considering to move it to Study area characteristic?
3.    In Introduction the aim of work must be clearly highlighted. Currently is very hard to conclude what is the aim of conducted work?
4.    What is the novelty of conducted analysis? What was Author’s motivation to undertaken this study. Also this information must be in Introduction.
5.    The name of chapter 2 should be “Study Area”. 
6.    Chapter 2.1. Please to provide more information about land use in analysed catchments and please specify more meteorological information (precipitation, temperature). 
7.    Table 2. The symbols of particular morphometric should be placed below the table. Also some symbols can cause confusion. E.g. the “Q” symbol in hydrology usually is used as “flow”. Better for altitude can be “H” symbol. The same with “P” symbol where in hydrology usually is using to describe precipitation. Please replace “hydrological network total length” with “river network total length”.
8.    I’m not sure if information in table 3 are necessary to study area characteristic.
9.    Chapter 2.2. The characteristic of Olbia city is too detailed. It should be significantly shortened. The Author’s should focus only on parameters which are important in point of view the urban flooding causing. The same comment for chapter 2.3. Livorno city.
10.    Chapter 3.1. In my opinion the name of chapter should be hydro-meteorological data. The Authors should briefly to describe the kind of data and their sources. Currently the description is too chaotic and difficult to read. 
11.    The research methodology should be first subchapter in Materials and Methods. In my opinion the table 4 is not necessary. It is enough if Authors mention the sources in main text. 
12.    Table 5: please provide the symbols below table.
13.    Generally the information in chapter 4.1. are described correctly but you can think about the simplification of description. For example the most important information, about rainfall event could be summarized in tables (rainfall duration, depth, intensity, date of occurrence and others). 
14.    In article I did not find the information about the course of flash floods, caused by analysed rainfall episodes. Because the Authors dispose detailed information about the rainfall, maybe worth considering re-creating the flood using rainfall-runoff models? Then we can obtain approximate information about flood (duration, peak flow, volume). Next the comparison with rainfall hyetograph can be made, what can inform about mutual relation between phenomena.

Author Response

At the first I would like to thank Authors for hard work and very interesting manuscript. 
I have read the paper carefully. The proposed topic is very important in hydrology because the urban flooding is still unsolved problem. In my opinion the presented article could be very interesting for researcher, who deal the similar problems.

Nevertheless I have some comments that I would like the Authors respond to.

  1.    In my opinion the Introduction must be re-written. Generally the Authors should be focused on flash floods. Currently there is too much general information about floods. The Authors must provide information about urban flooding (what are the main physiographic and meteorological factors causing them, how we can, how we can defend against them – rainfall water management, e.g.). 

R: In the introduction we have written about floods in general only in the first 10-11 lines. Then for 3 pages we have described only flash floods phenomena, from all points of view. We would like also to underline that the most cited papers about flash flood (Gaume et al, 2009 "A compilation of data on European flash floods" and then Marchi et al. 2010 "Characterisation of selected extreme flash floods in Europe and implications for flood risk management") are structurated in the same manner. In the introduction causes and factors are well described.

In spite of this, we have added a new specific part about urban flooding (lines 65-75).

  1.    Pages 3 and 4: in my opinion the table 1 is not necessary. It should be removed. The same figure 1. Maybe considering to move it to Study area characteristic?
    R: Thanks. Table 1 is very important because it highlights how the three Italian regions are highly prone to flash floods. We can reduce the table from 30 years to just 20 years (2000-2020). The same thing for figure 1 which has been reduced.

  2.    In Introduction the aim of work must be clearly highlighted. Currently is very hard to conclude what is the aim of conducted work?
    R: Thanks. We have added clearly the goal (lines 124-130).

  3.    What is the novelty of conducted analysis? What was Author’s motivation to undertaken this study. Also this information must be in Introduction.
    R: Thanks. We have added the novelty (lines 125-127).

    5.    The name of chapter 2 should be “Study Area”. 
    R: Thanks. We have modified the title.

  4.    Chapter 2.1. Please to provide more information about land use in analysed catchments and please specify more meteorological information (precipitation, temperature). 
    R: Thanks. We have added the requested information
  5.    Table 2. The symbols of particular morphometric should be placed below the table. Also some symbols can cause confusion. E.g. the “Q” symbol in hydrology usually is used as “flow”. Better for altitude can be “H” symbol. The same with “P” symbol where in hydrology usually is using to describe precipitation. Please replace “hydrological network total length” with “river network total length”.
    R: Thanks for the suggestion. The symbols have been changed.
  6.    I’m not sure if information in table 3 are necessary to study area characteristic.
    R: Thanks. Table 3 has been cancelled and transformed in the Annex n° 1.
  7.    Chapter 2.2. The characteristic of Olbia city is too detailed. It should be significantly shortened. The Author’s should focus only on parameters which are important in point of view the urban flooding causing. The same comment for chapter 2.3. Livorno city.
    R: Thanks. We have reduced both.
  8.    Chapter 3.1. In my opinion the name of chapter should be hydro-meteorological data. The Authors should briefly to describe the kind of data and their sources. Currently the description is too chaotic and difficult to read. 
    R: The title has been changed. The description is very easy to read: 8/10 lines each event. It is difficult to understand for us which is “chaotic”.
  9.    The research methodology should be first subchapter in Materials and Methods. In my opinion the table 4 is not necessary. It is enough if Authors mention the sources in main text. 
    R: Table n. 4 has been erased. Sources are described in the text. The research methodology now is the first subchapter in Materials and Methods, as you suggested.
  10.    Table 5: please provide the symbols below table.
    R: Sorry, but we have followed the guidelines instructions of LAND. The symbols must be placed above the table.
  11.    Generally the information in chapter 4.1. are described correctly but you can think about the simplification of description. For example the most important information, about rainfall event could be summarized in tables (rainfall duration, depth, intensity, date of occurrence and others). 
    R: Thanks. We are pleased that it is well described. All data are described extensively in the paragraph by images and graphics. The most important information have already been summarized in a table (now is n. 6) with the data and percentage.
  12.    In article I did not find the information about the course of flash floods, caused by analysed rainfall episodes. Because the Authors dispose detailed information about the rainfall, maybe worth considering re-creating the flood using rainfall-runoff models? Then we can obtain approximate information about flood (duration, peak flow, volume). Next the comparison with rainfall hyetograph can be made, what can inform about mutual relation between phenomena.

R: thanks for the suggestion. We are dealing with this aspect in an ongoing research, which will be the subject of a further publication: in detail we are developing volumetric rainfall-water discharge models aimed at evaluating the runoff coefficient for each case study (Livorno, Genoa and Olbia). We are also developing a comparison with the values obtained with the Curve Number method, also in the hypothetical case without urbanization.

Reviewer 3 Report

Faccini et al. compared the dynamics of three flood events that occurred in  three NW Italian coastal regions.

All maps require extreme geographic coordinates in the three panels.

Author Response

Faccini et al. compared the dynamics of three flood events that occurred in three NW Italian coastal regions.

All maps require extreme geographic coordinates in the three panels.

R: We decided to insert coordinates only in the first map, because the others are similar.

Round 2

Reviewer 2 Report

I would like to thank Authors for revised paper. All my concerns were solved. In my opinion the article meet the high requirements of Land and it can be publish in journal.